# A generalizable brain extraction net (BEN) for multimodal MRI data from rodents, nonhuman primates, and humans

**Ziqi Yu[1,2,3], Xiaoyang Han[1,2], Wenjing Xu[1,2], Jie Zhang[1,2], Carsten Marr[4], Dinggang Shen[5,6,7], Tingying Peng[8]\*, Xiao-Yong Zhang[1,2,3]\*, Jianfeng Feng[1,2,3]**

[1]Institute of Science and Technology for Brain-Inspired Intelligence, Fudan University, Shanghai, China; [2]MOE Key Laboratory of Computational Neuroscience and Brain-Inspired Intelligence, Fudan University, Shanghai, China; [3]MOE Frontiers Center for Brain Science, Fudan University, Shanghai, China; [4]Institute of AI for Health (AIH), Helmholtz Zentrum München, Neuherberg, Germany; [5]School of Biomedical Engineering, ShanghaiTech University, Shanghai, China; [6]Shanghai United Imaging Intelligence Co., Ltd, Shanghai, China; [7]Shanghai Clinical Research and Trial Center, Shanghai, China; [8]Helmholtz AI, Helmholtz Zentrum München, Neuherberg, Germany

**\*For correspondence:**
tingying.peng@helmholtz-muenchen.de (TP);
xiaoyong_zhang@fudan.edu.cn (X-YZ)

**Abstract** Accurate brain tissue extraction on magnetic resonance imaging (MRI) data is crucial for analyzing brain structure and function. While several conventional tools have been optimized to handle human brain data, there have been no generalizable methods to extract brain tissues for multimodal MRI data from rodents, nonhuman primates, and humans. Therefore, developing a flexible and generalizable method for extracting whole brain tissue across species would allow researchers to analyze and compare experiment results more efficiently. Here, we propose a domain-adaptive and semi-supervised deep neural network, named the Brain Extraction Net (BEN), to extract brain tissues across species, MRI modalities, and MR scanners. We have evaluated BEN on 18 independent datasets, including 783 rodent MRI scans, 246 nonhuman primate MRI scans, and 4601 human MRI scans, covering five species, four modalities, and six MR scanners with various magnetic field strengths. Compared to conventional toolboxes, the superiority of BEN is illustrated by its robustness, accuracy, and generalizability. Our proposed method not only provides a generalized solution for extracting brain tissue across species but also significantly improves the accuracy of atlas registration, thereby benefiting the downstream processing tasks. As a novel fully automated deep-learning method, BEN is designed as an open-source software to enable high-throughput processing of neuroimaging data across species in preclinical and clinical applications.

## Editor's evaluation

This article is an important contribution to the field of neuroimaging. The paper proposes a deep neural network for brain extraction and an approach to training the network that generalises across domains, including species, scanners, and MRI sequences. The authors provide convincing evidence that their approach works for a varied set of data, protocols, and species.

## Introduction

Cross-species studies from mice, nonhuman primates (NHPs) to humans are important for studying evolution, development, and the treatment of neurological disorders. As an ideal non-invasive imaging tool, magnetic resonance imaging (MRI) provides high-resolution, multimodal whole brain imaging of

**eLife digest** Magnetic resonance imaging (MRI) is an ideal way to obtain high-resolution images of the whole brain of rodents and primates (including humans) non-invasively. A critical step in processing MRI data is brain tissue extraction, which consists on removing the signal from the non-neural tissues around the brain, such as the skull or fat, from the images. If this step is done incorrectly, it can lead to images with signals that do not correspond to the brain, which can compromise downstream analysis, and lead to errors when comparing samples from similar species. Although several traditional toolboxes to perform brain extraction are available, most of them focus on human brains, and no standardized methods are available for other mammals, such as rodents and monkeys.

To bridge this gap, Yu et al. developed a computational method based on deep learning (a type of machine learning that imitates how humans learn certain types of information) named the Brain Extraction Net (BEN). BEN can extract brain tissues across species, MRI modalities, and scanners to provide a generalizable toolbox for neuroimaging using MRI. Next, Yu et al. demonstrated BEN's functionality in a large-scale experiment involving brain tissue extraction in eighteen different MRI datasets from different species. In these experiments, BEN was shown to improve the robustness and accuracy of processing brain magnetic resonance imaging data.

Brain tissue extraction is essential for MRI-based neuroimaging studies, so BEN can benefit both the neuroimaging and the neuroscience communities. Importantly, the tool is an open-source software, allowing other researchers to use it freely. Additionally, it is an extensible tool that allows users to provide their own data and pre-trained networks to further improve BEN's generalization. Yu et al. have also designed interfaces to support other popular neuroimaging processing pipelines and to directly deal with external datasets, enabling scientists to use it to extract brain tissue in their own experiments.

mice, NHPs, and humans. Brain tissue extraction on MRI data is a key processing step for the analysis of brain structure, function, and metabolism, as inaccurate brain extraction can not only lead to non-brain signal contamination or misleading registration, compromising the accuracy of downstream analyses, but also lead to cross-species comparison bias.

Currently, several brain extraction tools have been specifically designed for certain species or MRI modalities. These tools can be categorized into three groups based on their target species: (i) humans, (ii) NHPs, and (iii) rodents. Most well-established tools developed for humans, such as FreeSurfer (*Fischl, 2012*), Analysis of Functional NeuroImages (AFNI) (*Cox, 2012*), and the FMRIB Software Library (FSL) (*Jenkinson et al., 2012*), have been routinely used and integrated into standard preprocessing pipelines, for example, fMRIPrep (*Esteban et al., 2019*) and Data Processing and Analysis for Brain Imaging (DPABI) (*Yan et al., 2016*). In contrast, brain extraction in animals is far from being standardized or fully automated. Although a few atlas- or morphometry-based tools have been developed, for example, for NHPs (*Beare et al., 2013*; *Lohmeier et al., 2019*) and for rodents (*Chang et al., 2021*; *Liu et al., 2020*; *Nie and Shen, 2013*; *Oguz et al., 2014*), their performance is still limited.

Recently, deep learning (DL)-based algorithms have been developed to enable more accurate brain tissue segmentation in animals. For example, U-Net-based algorithms have been proposed to perform skull stripping automatically for NHPs (*Garcia-Saldivar et al., 2021*; *Wang et al., 2021*; *Zhao et al., 2018*; *Zhong et al., 2021*) and for rodents (*De Feo et al., 2021*; *Hsu et al., 2020*; *Valverde et al., 2020*). Although these DL-based approaches outperform traditional tools such as FreeSurfer or FSL in NHP and rodent brains, they have mostly been evaluated on a single species or a single MRI modality, and suffer severe performance degradation if they are applied to other species or modalities that differ from those represented in their training data. Despite the great diversity in neuroimaging studies in terms of species, MRI modalities, and platforms (e.g. different magnetic field strengths), accurately extracting brain tissue from images collected via multiple MRI modalities or platforms using a single tool remains very difficult, even without additionally attempting to cover various species comprising rodents, NHPs, and humans. In summary, a comprehensive and generalizable method to address these complex challenges is still lacking because current methods (1) are inflexible, which means that the available toolboxes are designed for certain species, modalities, or platforms and are

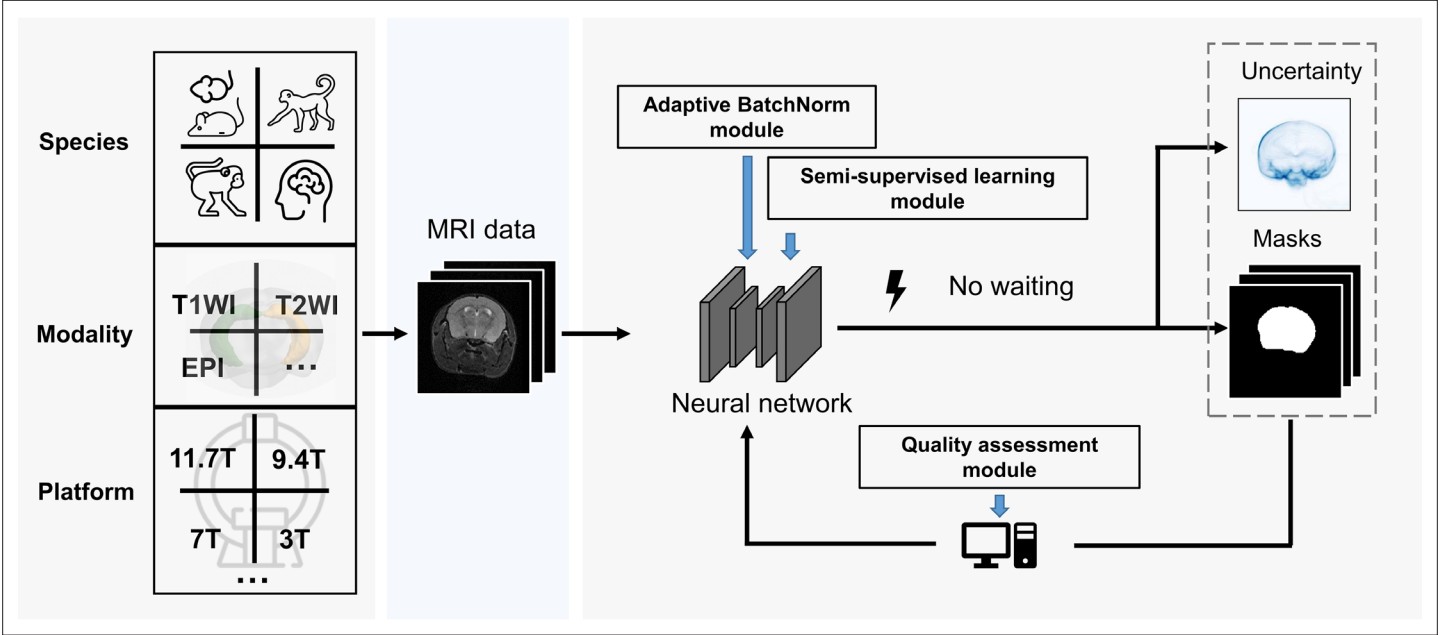

**Figure 1.** BEN renovates the brain extraction workflow to adapt to multiple species, modalities and platforms. The BEN has the following advantages: (1) Transferability and flexibility: BEN can adapt to different species, modalities and platforms through its adaptive batch normalization module and semi-supervised learning module. (2) Automatic quality assessment: Unlike traditional toolboxes, which rely on manual inspection to assess the brain extraction quality, BEN incorporates a quality assessment module to automatically evaluate its brain extraction performance. (3) Speed: As a DL-based method, BEN can process an MRI volume faster (<1 s) than traditional toolboxes (several minutes or longer).

not suitable for cross-species, cross-modality, or cross-platform application and (2) lack automatic quality assessment to evaluate the segmentation results, which means labor-intensive manual curation is required when erroneous segmentation occurs.

To address these issues, we propose a domain-adaptive and semi-supervised deep neural network, named the Brain Extraction Net (BEN, *Figure 1*), to perform the challenging task of accurately extracting brain tissues across different species (mouse, rat, marmoset, macaque, and human), across different MRI platforms with various magnetic field strengths (from 1.5T to 11.7T), and across different MRI modalities (structural MRI and functional MRI). Being flexible and generalizable, BEN is also considerably faster for brain extraction than using a traditional toolbox. Another feature of BEN is that it incorporates an uncertainty-based assessment module that allows it to autonomously control the quality of brain segmentation mask outputs. In contrast, when using a traditional toolbox, experts need to manually check hundreds of extracted brain scans which is very time-consuming. It is worth noting that this laborious manual inspection is not a one-time procedure; it must be repeated after making parameter changes when handling new data. We believe that an automated and generalizable brain extraction tool such as BEN can reduce intra- and interrater variability while also dramatically increasing throughput for the labor-intensive brain extraction task, hence facilitating many neuroimaging studies.

## Results
### Model design

In this study, we have collected 18 datasets (*Appendix 1—table 1*) to evaluate the generalizability of BEN, from different research institutions/cohorts worldwide, including 783 rodent MRI scans, 246 NHP MRI scans, and 4601 human MRI scans, totaling over 900,000 images. To the best of our knowledge, the datasets used in our study represent the largest-scale deep learning application of brain extraction reported to date in terms of the diversity in species, field strengths, and modalities.

Based on the complexity of our datasets, we designed a domain-adaptive and semi-supervised network (BEN, *Figure 2A*) to segment brain MRI volumes into brain and nonbrain tissues. The backbone of BEN is a U-shaped network with nonlocal attention architecture (NL-U-Net, *Figure 2B*). Unlike

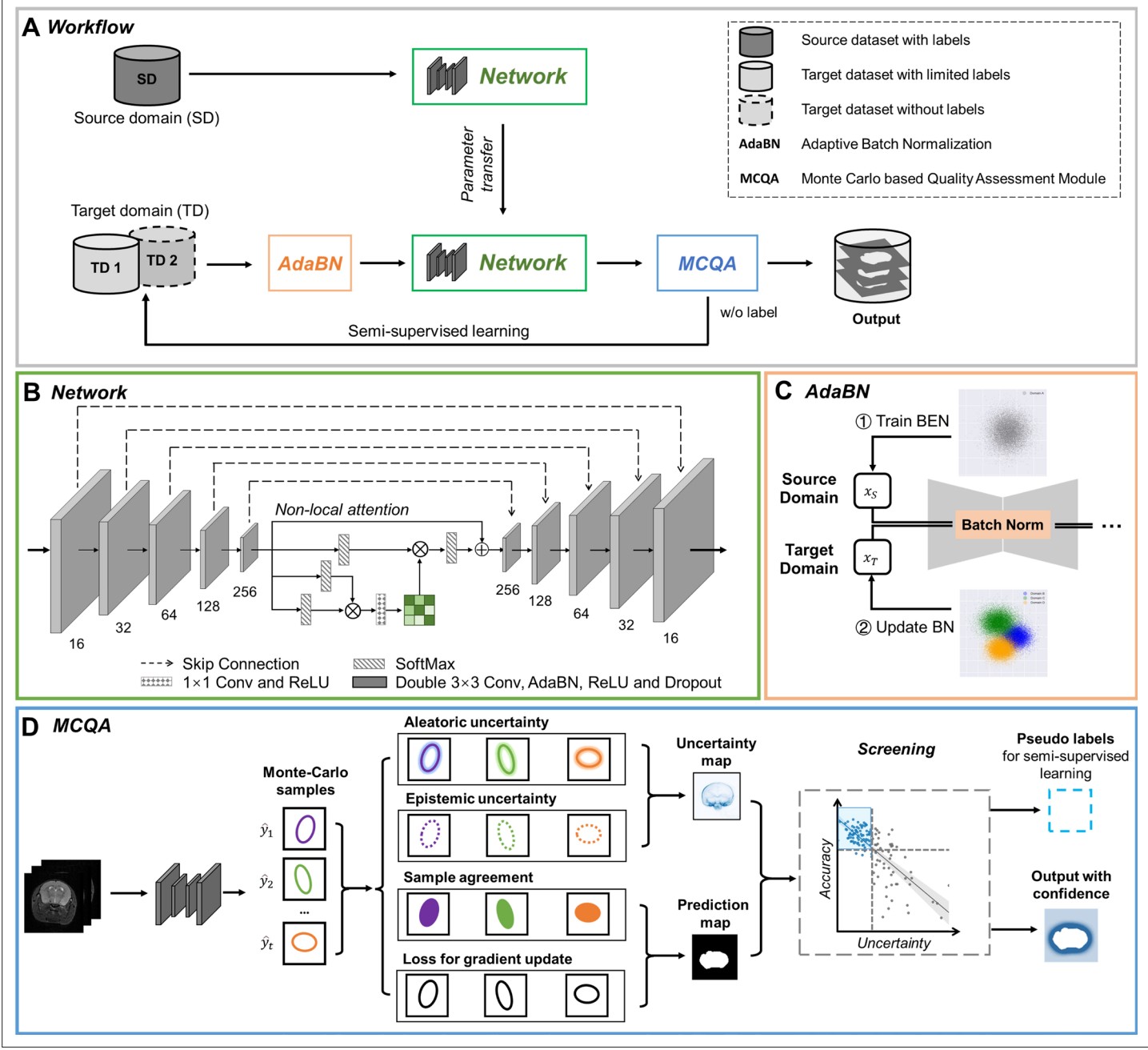

**Figure 2.** The architecture of our proposed BEN demonstrates its generalizability. (**A**) The domain transfer workflow. BEN is initially trained on the Mouse-T2-11.7T dataset (representing the source domain) and then transferred to many target domains that differ from the source domain in either the species, MRI modality, magnetic field strength, or some combination thereof. Efficient domain transfer is achieved via an adaptive batch normalization (AdaBN) strategy and a Monte Carlo quality assessment (MCQA) module. (**B**) The backbone of BEN is the nonlocal U-Net (NL-U-Net) used for brain extraction. Similar to the classic U-Net architecture, NL-U-Net also contains a symmetrical encoding and decoding path, with an additional nonlocal attention module to tell the network where to look, thus maximizing the capabilities of the model. (**C**) Illustration of the AdaBN strategy. The batch normalization (BN) layers in the network are first trained on the source domain. When transferring to a target domain, the statistical parameters in the BN layers are updated in accordance with the new data distribution in the target domain. (**D**) Illustration of the MCQA process. We use Monte Carlo dropout sampling during the inference phase to obtain multiple predictions in a stochastic fashion. The Monte Carlo predictions are then used to generate aleatoric and epistemic uncertainties that represent the confidence of the segmentations predicted by the network, and we screen out the optimal segmentations with minimal uncertainty in each batch and use them as pseudo-labels for semi-supervised learning.

previous approaches (*Garcia-Saldivar et al., 2021*; *Zhao et al., 2018*; *Zhong et al., 2021*; *De Feo et al., 2021*; *Hsu et al., 2020*; *Valverde et al., 2020*), the segmentation network of BEN is trained not from scratch but by leveraging domain transfer techniques achieved via two critical strategies: (i) an adaptive batch normalization (*Li et al., 2018*) (AdaBN, *Figure 2C*) strategy for adapting the statistical parameters of the batch normalization (BN) layers in a network trained on the source domain to the new data distribution of the target domain and (ii) an iterative pseudo-labeling procedure for semi-supervised learning in which a Monte Carlo quality assessment (MCQA, *Figure 2D*) method is proposed to assess the quality of the segmentations generated by the present network and to select the optimal pseudo-labels to be added to the training set for the next iteration. These two strategies are fully automatic and do not require any human intervention. After being trained on a source domain with abundant available annotations, BEN incorporates above domain transfer modules to achieve high segmentation accuracy in each target domain while requiring zero or only a limited number of target-domain annotations. For a detailed explanation of each model component, refer to the subsequent section.

## Experimental setup

Since T2WI is the most commonly used modality for rodent brain imaging and we have accumulated extensive brain scans with high-quality annotations in our previous research studies (*Han et al., 2021*; *Yu et al., 2021*), we first trained the model on the Mouse-T2WI-11.7T dataset, which served as the source-domain dataset, following the conventional fully supervised training strategy. Here, we used fivefold cross-validation with a training/testing split of 80%/20% (194/49 out of a total of 243 scans) to evaluate the performance of our trained model. The remaining 17 datasets serve as the target-domain datasets (one dataset corresponds to one domain). As shown in *Appendix 1—table 2*, BEN obtained a Dice score of 0.99 and a 95% Hausdorff distance (HD95) of 0.14 mm, outperforming other state-of-the-art (SOTA) approaches. The success of BEN in the source domain ensures that it has the potential to excel at a variety of downstream tasks.

For fair and comprehensive comparisons, we execute BEN and two benchmark settings, namely, training from scratch (training the model only on the target-domain dataset) and fine-tuning (first pretraining on the source-domain dataset and then adjusting the parameters for the target domain), on three domain transfer tasks: across species, across modalities and across MRI scanners with various magnetic fields. Among these three tasks, cross-species domain transfer is the most challenging because of the large anatomical and structural variations between different species.

## Transfer performance of BEN across species

We evaluated BEN's transferability from mouse MRI scans to scans from four other species: rat, marmoset, macaque and human. *Figure 3A–D* shows that the segmentation performance (quantified in terms of the Dice score and HD95) improves as the amount of labeled data increases in each of the four target species domains, as expected. In particular, BEN requires only two labeled rat MRI volumes, three labeled human volumes, and five labeled marmoset or macaque volumes to reach a Dice score of 95%, indicating satisfactory segmentation performance. In contrast, the two baseline methods, training from scratch and fine-tuning (*Hsu et al., 2020*; *Wang et al., 2021*), require a minimum of 10–15 labeled volumes in the rat, marmoset or macaque domain to reach the same threshold, i.e., almost five times as many labels as BEN. The only exception is human brain extraction, a relatively simple task compared to animal brain extraction, for which all three methods need only three labeled volumes to reach a Dice score of 95%. The more rapid progression to this performance threshold indicates a lower annotation cost, which is achieved by BEN through more efficient transfer learning. Additionally, visual assessment of several representative segmented brain volumes (*Figure 3E–H*) illustrates that BEN's segmentations are more consistent with the reference ground truth than those of other methods. For example, either training from scratch or fine-tuning often leads to false positive errors in the cranial and caudal regions of the brain, which could be caused by regional heterogeneity of the scans.

## Transfer performance across modalities

We further evaluated BEN's transferability across different MRI modalities, particularly between structural and functional modalities. Again, we used the structural Mouse-T2-11.7T dataset as the source

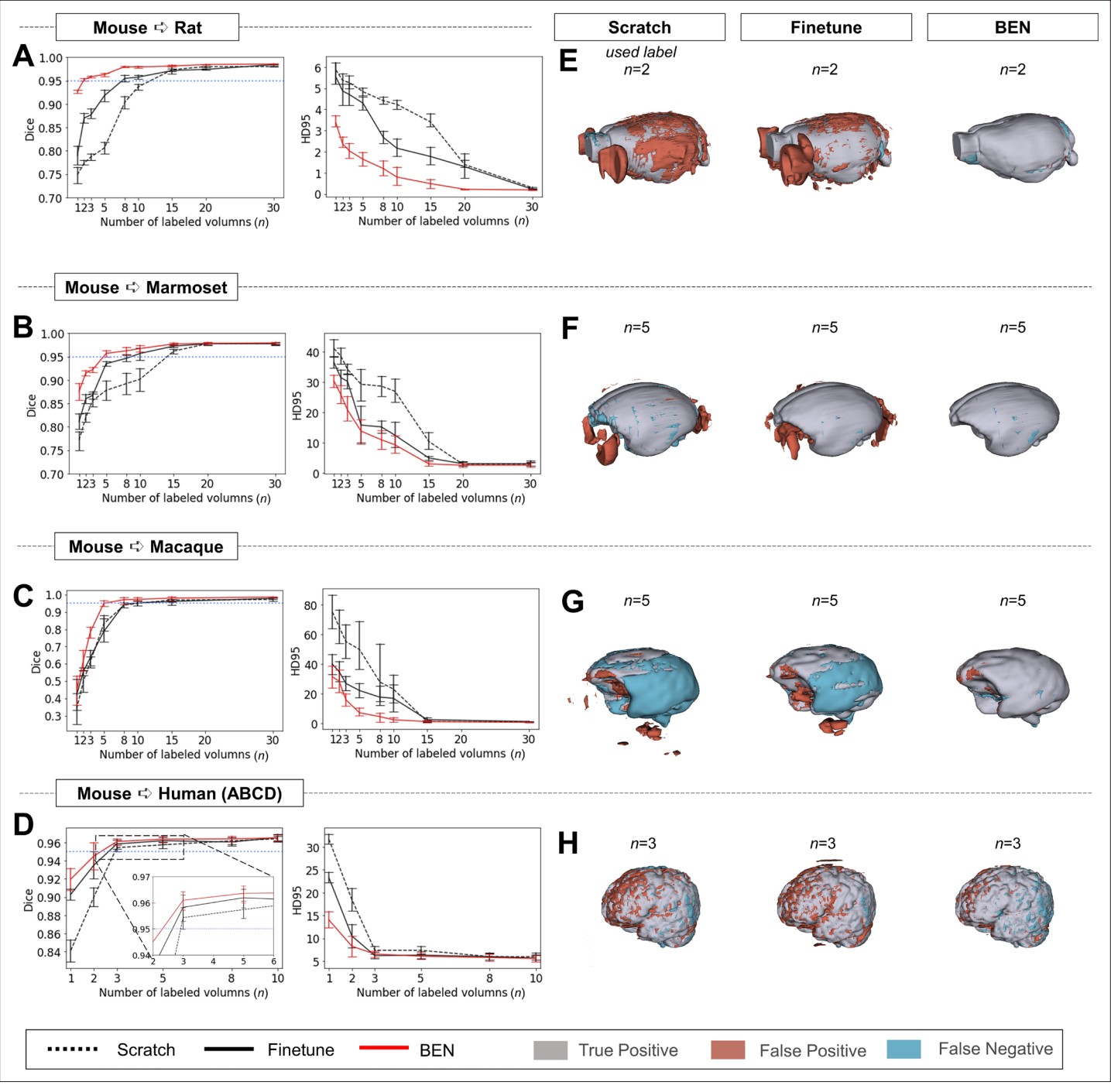

**Figure 3.** Performance comparison of BEN with two benchmark settings on the task of cross-species domain transfer. (**A - D**) Curve plots representing domain transfer tasks across species, showing the variation in the segmentation performance in terms of the Dice score and 95% Hausdorff distance (HD95) (y axis) as a function of the number of labeled volumes (x axis) for training from scratch (black dotted lines), fine-tuning (black solid lines) and BEN (red lines). From top to bottom: (**A**) mouse to rat, (**B**) mouse to marmoset, (**C**) mouse to macaque, and (**D**) mouse to human (ABCD) with an increasing amount of labeled training data (n=1, 2, 3, …, as indicated on the x axis in each panel). Both the Dice scores and the HD95 values of all three methods reach saturation when the number of labels is sufficient (n>20 labels); however, BEN outperforms the other methods, especially when limited labels are available (n≤5 labels). Error bars represent the mean with a 95% confidence interval (CI) for all curves. The blue dotted line corresponds to y(Dice)=0.95, which can be considered to represent qualified performance for brain extraction. (**E - H**) 3D renderings of representative segmentation results (the number (**n**) of labels used for each method is indicated in each panel). Images with fewer colored regions represent better segmentation results. (gray: true positive; brown: false positive; blue: false negative). Sample size for these datasets: Mouse-T2WI-11.7T (N=243), Rat-T2WI-11.7T (N=132), Marmoset-T2WI-9.4T (N=62), Macaque-T1WI (N=76) and Human-ABCD (N=963).

*Figure 3 continued on next page*

*Figure 3 continued*

The online version of this article includes the following figure supplement(s) for figure 3:

**Figure supplement 1.** Performance comparison of BEN with two benchmark settings on the task of cross-modality domain transfer.

**Figure supplement 2.** Performance comparison of BEN with two benchmark settings on the task of cross-platform domain transfer.

**Figure supplement 3.** BEN's transferability is not dependent on specific source dataset.

**Figure supplement 4.** UMAP visualizes BEN's transfer learning.

domain, and we transferred the trained model to three other commonly used MRI modalities: EPI, SWI and ASL (*Figure 3—figure supplement 1*). Encouragingly, we found that even without labels, BEN could generalize well, with Dice scores of 0.93–0.95 on all three target domains, despite the different imaging sequences and parameters between these modalities (*Figure 3—figure supplement 1A–C*). This is an impressive result, as zero-shot learning is generally a challenging problem (*Ma et al., 2021*; *Wang et al., 2019b*). By comparison, a fine-tuning strategy with zero labeled volumes, that is, a direct evaluation of the source-domain-trained model on the target domains, can achieve Dice scores of only 0.40–0.77, significantly lower than those of BEN (p<0.001 using the Mann–Whitney test). Note that training from scratch is impossible to implement without available labels in the target domain. Similar to the case of cross-species domain transfer, with fewer than five labeled volumes, BEN can achieve Dice >0.95 in the target domains in these modality transfer tasks (two volumes for T2WI to EPI, one volume for T2WI to SWI and five volumes for T2WI to ASL).

## Transfer performance across MR scanners with various magnetic field strengths

We also examined BEN's transferability to MRI datasets acquired from multiple MRI scanners with various magnetic field strengths (*Figure 3—figure supplement 2*). The obtained image contrast varies depending on the magnetic field strength. Even with the same magnetic field strength, MRI scanners from different manufacturers can produce different image contrast, which further increases the complexity of the domain shift. Without domain transfer techniques, the direct application of a model trained on a certain source domain to a different target domain strongly suffers due to the differences in imaging contrast/quality produced by the different scanners; specifically, Dice scores of only 0.42, 0.81, and 0.61 are obtained for the 11.7 T to 9.4 T, 11.7 T to 7 T, and 9.4 T to 7 T transfer tasks, respectively. In comparison, without additional labeled data in the target domains, BEN outperforms these benchmarks by notable margins, with Dice scores of 0.97, 0.95, and 0.91, respectively. Apparently, BEN shows the capacity to generalize well to other domains without additional labeled data, while the baseline approaches overfit the source domain. With sufficient labeled data in the target domain, all three methods perform well, as anticipated.

## Comparison with conventional neuroimage processing toolboxes

We compared BEN with several conventional SOTA toolboxes that are widely used in the neuroimaging research field, namely, AFNI, FSL, FreeSurfer, and Sherm (*Liu et al., 2020*), on all 18 datasets. Similar to the previous experiments, we used the Mouse-T2-11.7T dataset as the only source-domain dataset for BEN, with the remaining 17 datasets serving as the target-domain datasets. For each target domain, we used five labeled MRI volumes for the domain transfer task.

As shown in *Figure 4*, despite consistent performance on human MRI scans, the AFNI, FSL, and FreeSurfer toolboxes show large intra- and inter-dataset variations on the four animal species, suggesting that these methods are not generalizable to animals. Sherm, a toolbox that was specifically developed for rodent MRI, consequently achieves better results for mouse and rat scans but cannot be used for primates. In contrast, BEN is substantially superior to these conventional SOTA methods, and its performance remains consistent on both animal and human datasets, with median Dice scores of 0.97–0.99 for structural images (T2WI) (*Figure 4A–E*) and functional images (*Figure 4—figure supplement 1*) acquired at MRI scanners with various magnetic field strengths (*Figure 4*, *Figure 4—figure supplement 2*). Better brain segmentation leads to a more accurate estimation of the brain volume: as shown in *Figure 4F–J*, BEN achieves almost perfect linear regression coefficients (LRCs) of approximately 1.00 (red lines) when compared to the corresponding expert-labeled brain volumes. In contrast, AFNI, FSL and Sherm exhibit poor consistency in animals (LRC = 0.06–0.10 for mouse scans,

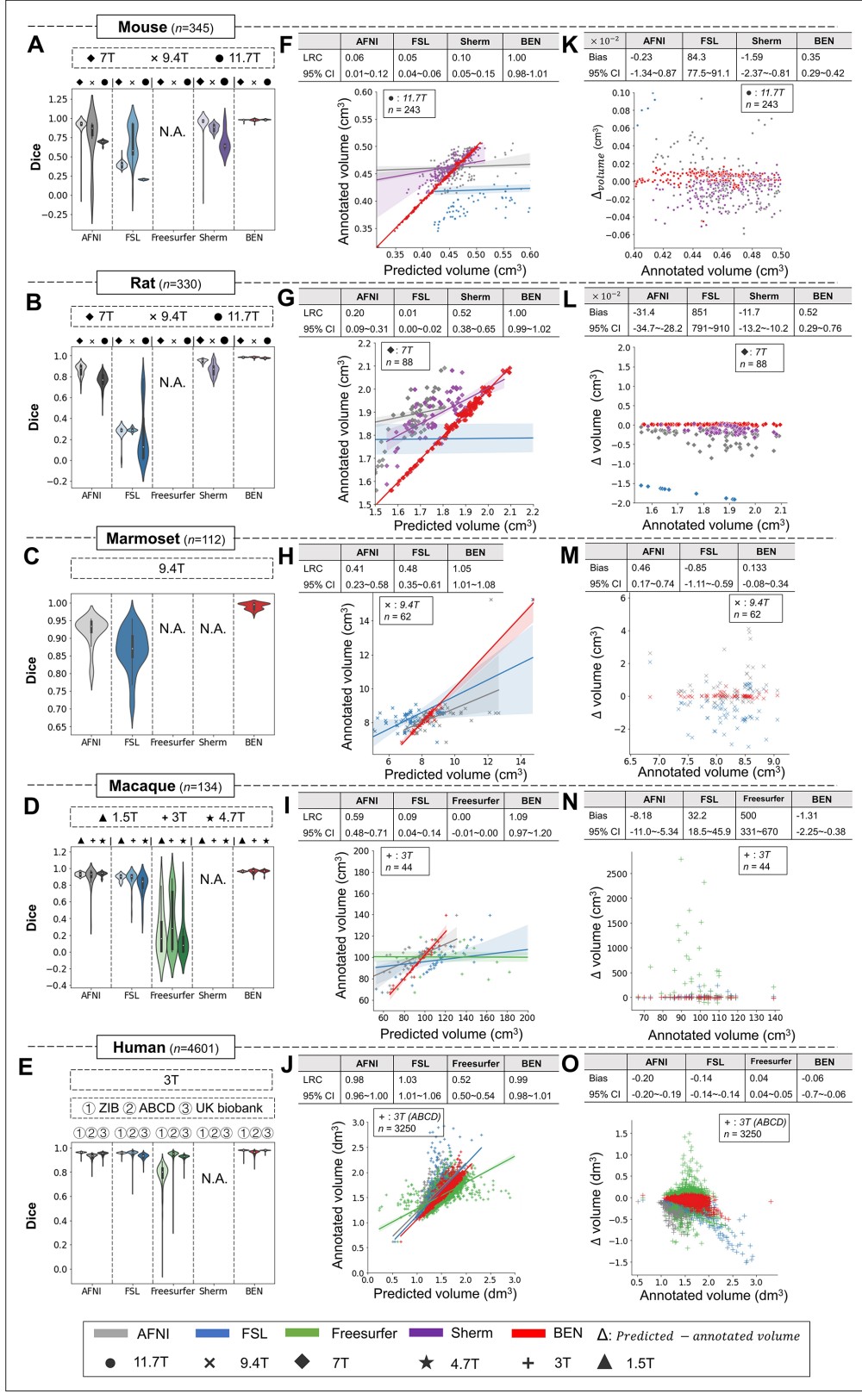

**Figure 4.** BEN outperforms traditional SOTA methods and advantageously adapts to datasets from various domains across multiple species, modalities, and field strengths. (**A - E**) Violin plots and inner box plots showing the Dice scores of each method for (**A**) mouse (n=345), (**B**) rat (n=330), (**C**) marmoset (n=112), (**D**) macaque (n=134), and (**E**) human (n=4601) MRI scans acquired with different magnetic field strengths. The field strength is illustrated

*Figure 4 continued on next page*

*Figure 4 continued*

with different markers above each panel, and the results for each method are shown in similar hues. The median values of the data are represented by the white hollow dots in the violin plots, the first and third quartiles are represented by the black boxes, and the interquartile range beyond 1.5 times the first and the third quartiles is represented by the black lines. 'N.A.' indicates a failure of the method on the corresponding dataset. (**F - J**) Comparisons of the volumetric segmentations obtained with each method relative to the ground truth for five species. For better visualization, we select magnetic field strengths of 11.7T for mouse scans (**F**), 7T for rat scans (**G**), 9.4T for marmoset scans (**H**), and 3T for both macaque (**I**) and human (**J**) scans. Plots for other field strengths can be found in *Figure 4—figure supplement 2*. The linear regression coefficients (LRCs) and 95% CIs are displayed above each graph. Each dot in a graph represents one sample in the dataset. The error bands in the plots represent the 95% CIs. (**K - O**) Bland–Altman analysis showing high consistency between the BEN results and expert annotations. The biases between two observers and the 95% CIs for the differences are shown in the tables above each plot. Δvolume = predicted volume *minus* annotated volume. Each method is shown using the same hue in all plots (gray: AFNI; blue: FSL; green: FreeSurfer; purple: Sherm; red: BEN). Different field strengths are shown with different markers (●: 11.7T; ×: 9.4T; ◆: 7T; ★: 4.7T; +: 3T; ▲: 1.5T). The Dice scores compute the overlap between the segmentation results and manual annotations.

The online version of this article includes the following figure supplement(s) for figure 4:

**Figure supplement 1.** BEN outperforms traditional SOTA methods in functional MRI scans acquired from multiple species on different magnetic field strengths.

**Figure supplement 2.** Linear regression coefficients and Bland–Altman analysis demonstration of the rest datasets.

**Figure supplement 3.** Error maps of BEN and SOTA methods.

**Figure supplement 4.** Execution time comparison of BEN with other methods.

---

0.01–0.52 for rat scans, 0.41–0.48 for marmoset scans and 0.00–0.59 for macaque scans). Additionally, in Bland–Altman plots (*Figure 4K–N*), BEN shows higher agreement between the predicted and manually annotated volumes than the other toolboxes. The error maps further confirm that BEN has better performance compared to other methods (*Figure 4—figure supplement 3*). The only exception is for human MRI data, with AFNI and FSL achieving excellent agreement with the ground truth on three human datasets, namely, the ABCD, UK Biobank and ZIB datasets (with LRC around 1.00). This is not surprising, as these conventional tools are well designed for human brain extraction. Encouragingly, BEN also exhibits comparable or even better performance for the human brain (*Figure 4E, J, O*), with an accelerated speed of 0.6 s per scan on average. This speed is two orders of magnitude greater than that of conventional toolboxes (several minutes on average) and three orders of magnitude greater than that of manual annotation (an average of 20–30 min; *Figure 4—figure supplement 4*), suggesting the potential of applying BEN in high-throughput studies. Taking into consideration that in practice, UK Biobank and Human Connectome Project (HCP) use more time-consuming registration-based approaches in their preprocess pipelines, which further emphasize the requirement for a fast and robust brain extraction toolbox.

In addition, we also compared BEN with published DL-based SOTA brain extraction methods (*Chou et al., 2011*; *Hsu et al., 2020*; *Wang et al., 2021*) as well as non DL-based methods (RATS (*Oguz et al., 2014*), Sherm (*Liu et al., 2020*), AFNI, FSL, and FreeSurfer), on two public datasets, CARMI (https://openneuro.org/datasets/ds002870/versions/1.0.0) and PRIME-DE (https://fcon_1000.projects.nitrc.org/indi/indiPRIME.html). As shown in *Appendix 1—table 3*, the segmentation performance of BEN, as evaluated using Dice score and the Hausdorff distance, surpasses other methods.

## BEN improves the atlas-to-target registration and brain volumetric quantification

Accurate brain extraction is a key component of many neuroimaging pipelines, facilitating subsequent procedures in longitudinal or cohort analyses. We therefore designed experiments to demonstrate BEN's application for downstream processing and analysis in neuroimaging studies of rodents and humans (*Figure 5* and *Figure 5—figure supplement 1*), including atlas registration (*Figure 5*) and volumetric quantification (*Figure 5—figure supplement 1*). The registration quality was evaluated in terms of the Dice scores between the brain structural segmentations of the warped MRI brain images and the atlas. *Figure 5B, F, J* plot the workflow of atlas registration, in which mice are registered to

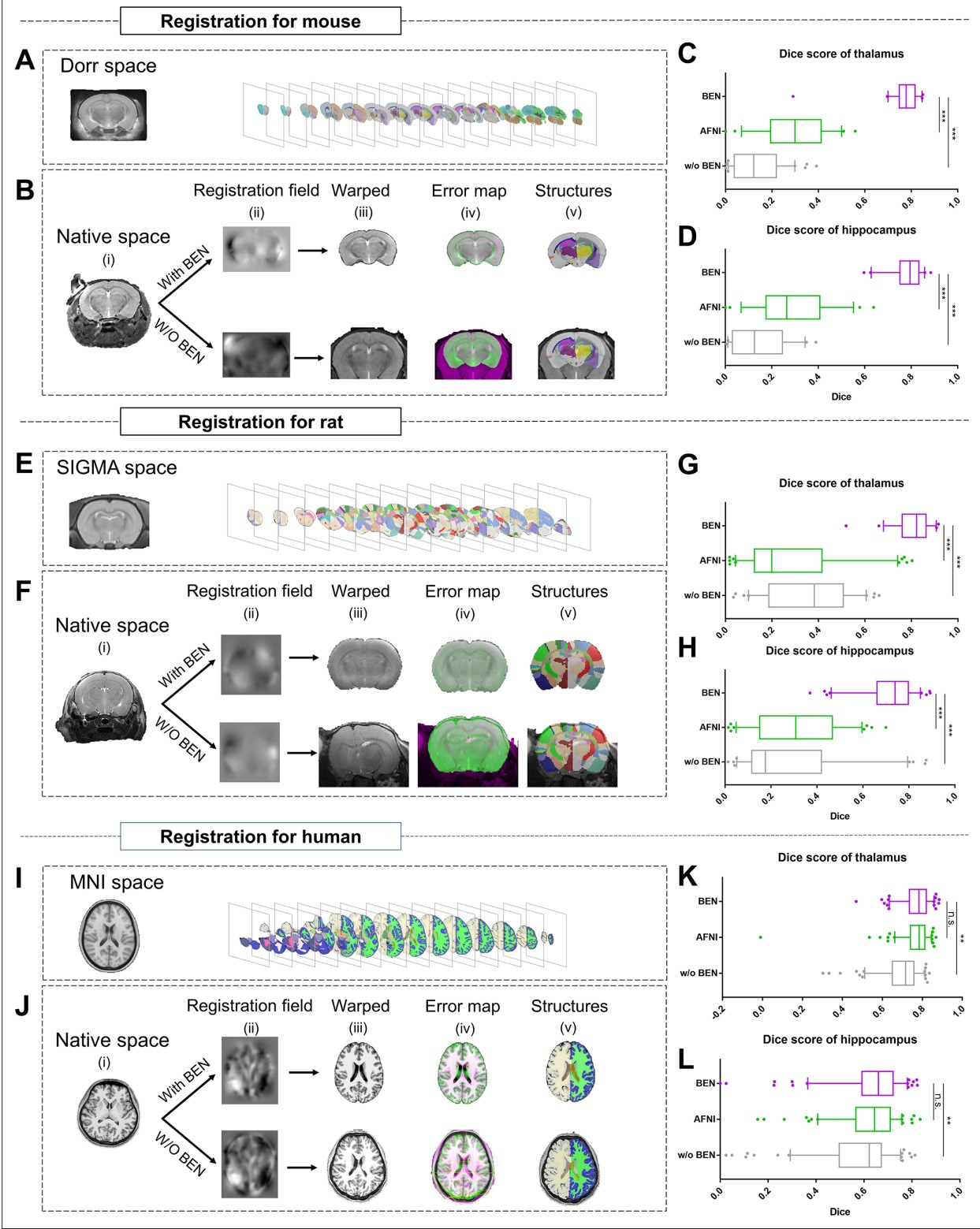

**Figure 5.** BEN improves the accuracy of atlas registration by producing high-quality brain extraction results. (**A**) The Dorr space, (**E**) the SIGMA space and (**I**) the MNI space, corresponding to the mouse, rat and human atlases, respectively. (**B, F, J**) Integration of BEN into the registration workflow: (i) Three representative samples from a mouse (n=157), a rat (n=88), and human (n=144) in the native space. (ii) The BEN-segmented brain MRI volumes in the native space as registered into the Dorr/SIGMA/MNI space using the Advanced Normalization Tools (ANTs) toolbox, for comparison with the registration of AFNI-segmented/original MRI volumes with respect to the corresponding atlas. (iii) The warped volumes in the Dorr/SIGMA/MNI spaces.

*Figure 5 continued on next page*

*Figure 5 continued*

(iv) Error maps showing the fixed atlas in green and the moving warped volumes in magenta. The common areas where the two volumes are similar in intensity are shown in gray. (**v**) The brain structures in the warped volumes shown in the atlas spaces. In our experiment, BEN significantly improves the alignment between the propagated annotations and the atlas, as confirmed by the improved Dice scores in the (**C, G, K**) thalamic and (**D, H, L**) hippocampal regions (box plots: purple for BEN, green for w/o BEN; volumes: n=157 for the mouse, n=88 for the rat, n=144 for the human; statistics: paired t-test, n.s.: no significance, *p<0.05, **p<0.01, ***p<0.001).

The online version of this article includes the following figure supplement(s) for figure 5:

**Figure supplement 1.** Atlas registration with BEN benefits brain volumetric quantification in longitudinal studies.

Dorr (*Dorr et al., 2008*), rats to SIGMA (*Barrière et al., 2019*), and humans to MNI (*Evans et al., 2005*). Quantitative evaluations of two critical brain structures, the thalamus and hippocampus (*Figure 5C, D, G, H, K, L*), demonstrate that BEN can tremendously enhance the registration quality; registration with AFNI achieves a Dice score of only 0.2–0.6, which can be improved to approximately 0.8 when using BEN for brain extraction (p<0.001).

As the next step, we examined the contribution of BEN to brain volumetric quantification in rodent longitudinal studies. Two longitudinal MRI datasets, representing adult mice (8, 12, 20, and 32 weeks old, *Figure 5—figure supplement 1A*) and adolescent rats (3, 6, 9, and 12 weeks old, *Figure 5—figure supplement 1C*), were collected. As shown in *Figure 5—figure supplement 1B*, after registration with BEN, atlas-based morphometry analysis showed that the volumes of two representative brain regions, the thalamus and hippocampus, remained stable for the adult mice from 8 weeks old to 32 weeks old (purple lines). In contrast, calculating the volumes of these brain regions without BEN led to unreliable quantifications (green lines) due to poor atlas registration resulting in the propagation of error to the volumetric quantifications. Similarly, BEN improved brain volumetric quantification in adolescent rats and yielded a plausible pattern of growth from 3 weeks old to 12 weeks old (*Figure 5—figure supplement 1D*, purple boxes), suggesting rapid development of these two critical brain regions during this period, consistent with findings in the literature (*Calabrese et al., 2013*; *Hamezah et al., 2017*). In comparison, the atlas-based segmentations were poor when without BEN, and no obvious trend in the volumetric statistics was apparent (green boxes). These results indicate that BEN not only is able to tremendously improve the registration accuracy and contribute to routine brain MRI processing but also is critical for longitudinal MRI studies, as it improves volumetric quantification.

## BEN represents interrater variations in the form of uncertainty maps

Due to different levels of annotation experience, disagreements or disputes between human annotators remain a problem. To quantitatively assess the bias of interrater disagreements, we carried out a case study of the annotation of brain functional MRI images from mice, rats, marmosets, and macaques. A total of 9 raters participated in the experiments: two senior raters (defined as having >5 years of experience) and seven junior raters (defined as having 1–2 years of experience). The consensus region of the two senior raters' annotations was taken as the ground truth (reference) for comparison with the annotations of the seven junior raters. BEN generated an uncertainty map through Monte Carlo sampling during the inference process, without additional adjustment of the network structure.

As shown in *Figure 6*, the segmentations produced by BEN match the ground truth very well across different species. The attention maps and uncertainty maps display complementary information about the brain morphology; specifically, the attention maps focus on the image signal within the brain tissue in MRI images (*Figure 6C*), while the uncertainty maps provide information on the boundaries between the brain and nonbrain tissues (*Figure 6D*). Interestingly, for each rater's annotations, disputes always occur at the brain boundaries (*Figure 6E*), similar to BEN's uncertainty map. The concurrence between the projections of BEN's uncertainty maps and the rater disagreement maps (*Figure 6—figure supplement 1*) further demonstrates that BEN has the potential to interrogate its own predictions and that its uncertainty measure can be deemed an alternative proxy for rater divergence. From an intra-observer perspective, the nature of the slice-wise approach potentially impacts the consistency of attention maps for human scans with isotropic spatial resolution. One ideal solution is to extend BEN to the 2.5D method or use 3D convolution kernels for these imaging cohorts.

Furthermore, to evaluate the annotation variability between different raters in a more quantitative fashion, we calculated the Dice scores and found that none of them was higher than 0.96 across the

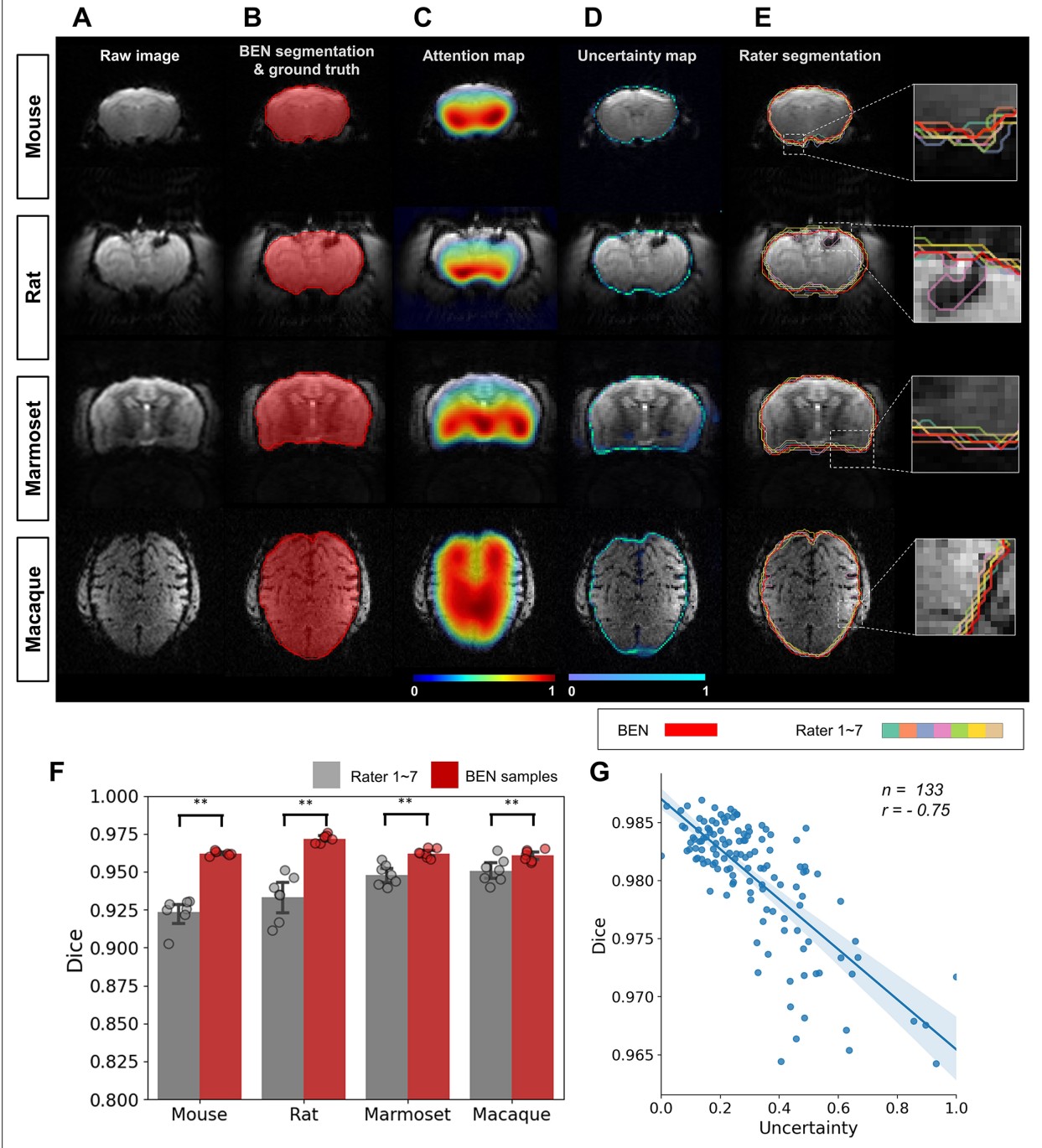

**Figure 6.** BEN provides a measure of uncertainty that potentially reflects rater disagreement. (**A**) Representative EPI images from four species (in each column from left to right, mouse, rat, marmoset, and macaque). (**B**) BEN segmentations (red semi-transparent areas) overlaid with expert consensus annotations (red opaque lines), where the latter are considered the ground truth. (**C**) Attention maps showing the key semantic features in images as captured by BEN. (**D**) Uncertainty maps showing the regions where BEN has less confidence. The uncertainty values are normalized (0–1) for better visualization. (**E**) Annotations by junior raters (n=7) shown as opaque lines of different colors. (**F**) Bar plots showing the Dice score comparisons between the ground truth and the BEN segmentation results (red) as well as the ground truth and the junior raters' annotations (gray) for all species (gray: raters, n=7, red: Monte Carlo samples from BEN, n=7; statistics: Mann–Whitney test, **p<0.01). Each dot represents one rater or one sample. Values are represented as the mean and 95% CI. (**G**) Correlation of the linear regression plot between the Dice score and the normalized uncertainty. The error band represents the 95% CI. Each dot represents one volume (species: mouse; modality: T2WI; field strength: 11.7T; n=133; r=−0.75).

The online version of this article includes the following figure supplement(s) for figure 6:

**Figure supplement 1.** BEN's uncertainty is consistent with interrater' disagreement.

*Figure 6 continued on next page*

*Figure 6 continued*

**Figure supplement 2.** Interrater disagreement.

**Figure supplement 3.** BEN provides a measure of uncertainty that potentially reflects the disagreement of conventional toolboxes in human data.

four species (*Figure 6—figure supplement 2*). Specifically, the Dice scores between each junior rater and the ground truth provided by the senior raters are between 0.92 and 0.94 for the four species, and these are surpassed by BEN, with Dice scores of 0.96–0.97 (p<0.01) (*Figure 6F*). In addition to animal studies, we performed a parallel experiment to verify BEN's predictions for human brain MRI. As shown in *Figure 6—figure supplement 3*, similar to the animal segmentation results, BEN produces attention and uncertainty maps to support interpretation of BEN's robustness compared with conventional toolboxes.

We further examined the correlation between BEN's uncertainty and segmentation quality. As shown in *Figure 6G*, the estimates of uncertainty at the volume level are negatively correlated with the Dice score ($r=-0.75$), suggesting that BEN's uncertainty measure can be considered as an alternative metric for assessing the quality of segmentations at the time of inference, when the Dice score cannot be computed because the ground-truth segmentation is not available.

## Discussion

In this study, we introduce BEN, a generalized brain extraction tool for MRI data from rodents, NHPs, and humans. Our results demonstrate BEN's transferability across different species, modalities, and platforms thanks to several key modules: an AdaBN module, a semi-supervised learning module, and an uncertainty estimation module (the contribution of each module refers to ablation study in *Appendix 1—tables 4 and 5*). To facilitate BEN's usage, we also provide interfaces to make BEN compatible with several popular toolboxes, including AFNI, FSL, and FreeSurfer, and therefore easy to integrate into conventional neuroimage processing pipelines.

### Transferability

By virtue of its modules for domain adaptation and transfer learning, BEN is capable of handling the heterogeneity in MRI contrast associated with different subjects, MRI scanners, and imaging sequences. It is worth noting that the source domain for BEN can be freely chosen. In this study, we switched the source domain for BEN's deployment from a mouse dataset to a human dataset (*Figure 3—figure supplement 3*). Similar to the successful transfer of BEN from mouse data to human data (Dice scores of 0.96 with only 3 labeled scans, *Figure 3*), BEN can also achieve an impressive Dice score higher than 0.95 when only 1 labeled volume is used for transfer from human data to mouse data. This dual transferability between mouse and human data demonstrates that BEN can take advantage of the semantic features extracted from one domain and transfer the corresponding learned information to the other domain despite the enormous gap in brain morphometry between the two species. This also indicates that the source domain could be represented by any brain MRI dataset from any other species. On the project website, we have released trained model weights for five species (mouse, rat, marmoset, macaque, and human) to allow users to start domain transfer from the closest species to those of interest to them with minimal or even zero additional annotations.

To interpret how transfer learning affects network performance, we used uniform manifold approximation and projection (UMAP) (*McInnes et al., 2018*) to visualize the feature distributions before and after domain transfer (details are described in the Materials and methods section). *Figure 3—figure supplement 4* shows a representative cross-modality transfer task from Mouse-T2WI-11.7T to Mouse-EPI-11.7T, which shows substantial perceptual differences. Although the semantic features are separated by the network in the source domain (*Figure 3—figure supplement 4A*), they are intermingled in the target domain without transfer learning (*Figure 3—figure supplement 4B*). In contrast, only after BEN's transfer modules are those features again separated in the target domain (*Figure 3—figure supplement 4C*), demonstrating the importance of using the domain transfer strategy.

### Automatic quality assessment and 'virtual' annotation

Just as divergences always exist among human annotators, the quality of neural-network-generated segmentations may also vary greatly among diverse sample data due to the inherent data distribution

or noise (*Baumgartner et al., 2019*; *Hüllermeier and Waegeman, 2021*; *Zeng et al., 2021*). However, the uncertainty of a network's decisions regarding its generated segmentations can be assessed via Monte Carlo sampling in many medical image analyses (*Jungo et al., 2018*; *Mehrtash et al., 2020*; *Wang et al., 2019a*). This is also the basis of BEN's uncertainty assessment module. Our results demonstrate that BEN's uncertainty map represents the expected interrater variations and that the uncertainty linearly increases as BEN's performance decreases. Based on the previously presented rater experiment (*Figure 6*), BEN generates segmentations that are comparable to the consensus segmentations of two senior experts with high consistency, indicating that BEN can apply consistent labeling rules alongside automatic quality assessments when constructing large datasets to prevent subjective bias in large-scale research.

In addition, BEN is able to produce uncertainty-guided pseudo-labels with high confidence by using a semi-supervised learning strategy. Note that on the ABCD human dataset, BEN achieves competitive or even superior performance with fewer than 5 labels, nearly perfectly delineating thousands of scans (n=3250; *Figure 4E and J & O*); this further demonstrates that BEN can propagate annotations from limited labeled data to unlabeled data and produce a larger set of 'virtual' labels for training, suggesting that it can be easily applied to large-scale datasets with limited annotations for rapid automatic deployment. Nevertheless, one limitation of our model is that BEN's quality assessment may result in deviations when the MRI images are of poor quality, such as motion artifacts and low signal-to-noise ratios (SNRs). In future work, this procedure may be updated by incorporating conditional random fields (*Xie et al., 2021*) or heuristic algorithms.

## Compatibility with other toolboxes and deployment

BEN was developed in Python and is equipped with well-designed interfaces to support most types of software encountered in existing traditional pipelines (*Appendix 1—table 6*), including FSL, AFNI, FreeSurfer, Advanced Normalization Tools (ANTs), and statistical parametric mapping (SPM). We also provide tutorials and demo examples to flatten the learning curve and lower the entry barrier for researchers deploying our new algorithm. Moreover, BEN was built under the open-source paradigm from the beginning and has already been tested on various types of MRI data from many imaging centers. We have released packaged interfaces for BEN and source codes, including the implementations for domain transfer, semi-supervised training and quality control. Additionally, we have distributed the model weights trained on five species (mouse, rat, marmoset, macaque and human). Applications to other target domains can also benefit from our pretrained models, requiring the addition of zero or only a small number of labeled images to update the model.

For the introduction of a new dataset, we suggest that the user do as follows. (1) First, use BEN to load the pretrained weights for the closest corresponding species. (2) Run unsupervised domain adaptation on the user's dataset. The customized weight will be updated and saved automatically, and the user will then use it to execute BEN on their data. (3) Consider labeling several scans to fine-tune the model if the self-configuring weights do not yield satisfactory performance, or the target species is beyond the scope of our datasets. The domain transfer procedures will be performed automatically, without human intervention. (4) If BEN is deployed to an external image domain/cohort, retraining the BEN might be required. In addition, there are two options to improve BEN's generalizability. (1) The scans selected for retraining or fine-tuning BEN should be representative of target cohorts, taking into consideration field bias and brain injury (if any). (2) Pre- and post-processing steps are simple to execute yet effective, and these could be easily integrated into the BEN pipeline. For example, we have already provided largest connected region selection, orientation detection, conditional random field optimization, etc. as plug-and-play functions in our pipeline.

When dealing with external datasets, there could be a couple of reasons that cause suboptimal performance using a pretrained BEN. On the one hand, domain generalization is a challenging task for deep learning. Although BEN could adapt to new out-of-domain images without labels (zero-shot learning) when the domain shift is relatively small (e.g. successful transfer between modalities and scanners with different MR strengths), the domain gap existing in different cohorts might compromise the performance. In this case, additional labeled data and retraining are indeed necessary for BEN to perform few-shot learning. On the other hand, users can provide their own data and pretrained network as a new source domain to take advantage of BEN's flexibility, which does not bind to a fixed

source domain, therefore facilitating domain generalization by reducing the domain gap between the new source and target domains.

BEN is designed as an extensible toolbox and follows the open-access paradigm, allowing users to save their updated models and share their weights for use by the neuroimaging community. The accumulation of additional imaging data will further improve the performance and generalization of BEN and support the exploration of complex neuroimaging research.

## Conclusion

In summary, we have demonstrated the superiority of BEN in terms of accuracy, robustness and generalizability in a large-scale study involving eighteen different MRI datasets. BEN improves the robustness of atlas-to-target registration and brain volumetric quantification in neuroimaging. BEN is an open-source and modularly designed toolbox that provides uncertainty maps to quantify the confidence of the network's decisions and to model interrater variability. We believe that BEN has great potential to enhance neuroimaging studies at high throughput for both preclinical and clinical applications.

# Materials and methods

## Datasets

We conducted experiments to evaluate BEN on eighteen datasets covering five distinct species, namely, mouse, rat, marmoset, macaque, and human; four MRI modalities, namely, T1- and T2-weighted imaging (T1WI/T2WI), echo-planar imaging (EPI), susceptibility-weighted imaging (SWI) and arterial spin labeling (ASL); and six MRI platforms spanning a wide range of magnetic field strengths, namely, 1.5T, 3T, 4.7T, 7T, 9.4T, and 11.7T. Each dataset contains 14–3250 scans acquired from different research institutions/cohorts worldwide, including China, the United States, the United Kingdom, France, Canada, and the Netherlands. Partial rodent MRI data collection were approved by the Animal Care and Use Committee of Fudan University, China. The rest rodent data (Rat-T2WI-9.4T and Rat-EPI-9.4T datasets) are publicly available (CARMI: https://openneuro.org/datasets/ds002870/versions/1.0.0). Marmoset MRI data collection were approved by the Animal Care and Use Committee of the Institute of Neuroscience, Chinese Academy of Sciences, China. Macaque MRI data are publicly available from the nonhuman PRIMatE Data Exchange (PRIME-DE) (https://fcon_1000.projects.nitrc.org/indi/indiPRIME.html; **Milham et al., 2018**). The Zhangjiang International Brain Biobank (ZIB) protocols were approved by the Ethics Committee of Fudan University (AF/SC-03/20200722) and written informed consents were obtained from all volunteers. UK Biobank (UKB) and Adolescent Brain Cognitive Development (ABCD) are publicly available. Detailed information on each dataset is shown in *Appendix 1—table 1* and summarized below:

1. Mouse: includes seven datasets consisting of 453 scans (14,358 slices), namely, Mouse-T2WI-11.7T (243 scans), Mouse-T2WI-9.4T (14 scans), Mouse-T2WI-7T (14 scans), Mouse-EPI-11.7T (54 scans), Mouse-EPI-9.4T (20 scans), Mouse-SWI-11.7T (50 scans), and Mouse-ASL-11.7T (58 scans).
2. Rat: includes four datasets consisting of 330 scans (11,264 slices), namely, Rat-T2WI-11.7T (132 scans), Rat-T2WI-9.4T (55 scans), Rat-T2WI-7T (88 scans) and Rat-EPI-9.4T (55 scans).
3. Marmoset: includes two datasets consisting of 112 scans (4,060 slices), namely, Marmoset-T2WI-9.4T (65 scans) and Marmoset-EPI-9.4T (50 scans).
4. Macaque: includes two datasets consisting of 134 scans (22,620 slices), namely, Macaque-T1WI (76 scans) and Macaque-EPI (58 scans).
5. Human: includes three datasets consisting of 4,601 scans (87,3453 slices): Human-ABCD (3,250 scans), Human-UKB (963 scans) and Human-ZIB (388 scans).

## Backbone network architecture

The backbone of BEN is based on a U-shaped architecture (*Ronneberger et al., 2015*), as is commonly used for biomedical volumetric segmentation (*Isensee et al., 2021*). We use 2D convolution kernels rather than 3D kernels here to accelerate the execution speed and reduce the demand for GPU memory. In addition, most rodent and NHP scans have an anisotropic spatial resolution, so 3D convolution is not appropriate. As shown in *Figure 2B*, the backbone network consists of five

symmetrical pairs of encoding and decoding blocks, which perform contraction and expansion operations, respectively, during the data feedforward process. Each encoding/decoding block is composed of two convolutional layers with 3*3 convolution kernels, followed by a BN layer, an exponential linear unit (ELU) function and a 2*2 max-pooling operation with stride 2. The first encoding block results in 16 feature channels, and every subsequent encoding block doubles the number of channels. Each decoding block is symmetric with respect to the encoding block at the corresponding level, except that the max pooling layer is replaced with transposed convolutional layers to upsample the feature map. In the last decoding block, we add a 1*1 convolutional layer with a 3*3 kernel size and a sigmoid activation function to map the 16 feature channels to a binary probability map that softly assigns each pixel to the brain and nonbrain tissue classes.

Additionally, we add a self-attention mechanism in the bottom layer of the U-Net to yield higher accuracy and smoother boundaries. A self-attention block (*Wang et al., 2017*) is used to improve the model performance for various domain tasks. Concretely, we use a nonlocal attention layer in the bottleneck of the network to model long-range and global contextual dependencies over feature representations. This attention mechanism calculates a weighted saliency map at each position in the feature map that accounts for the global contextual response, thus better assisting the network in distinguishing brain and nonbrain tissue in specific domains with different contrast characteristics and distributions.

The model takes preprocessed MRI images as input (for details, refer to the subsection titled 'Model input preprocessing') and merges successive mask slices to reconstruct the original 3D volumes. Here, we mainly focus on the structure and components of the model as well as the training and evaluation procedures in the source domain. The pretrained model weights will be retained for the transfer learning and domain adaptation stages, which are discussed in the next section.

## Adaptive batch normalization (AdaBN)

To mitigate the domain shift problem encountered when DL models are applied to different domains, we adopt the AdaBN strategy (*Li et al., 2018*). Standard BN layers use fixed statistical descriptors for multiple domains, thus damaging the generalization ability of the trained model. In contrast, in AdaBN, the statistical descriptors are updated in accordance with the specific target domain during deployment, resulting in better model generalization. Concretely, when deployed to target domains, we first freeze all network layers other than batch normalization layers, and then perform forward propagation using target domain data without labels being present. The differences in intensity distributions that exist between the source and target domains will be captured via batch normalization layers, and biased statistical descriptors (mean and variance) will be corrected. For our brain segmentation task, AdaBN can help the model to automatically adapt to the different MRI intensity distributions associated with different species, modalities and platforms without requiring modifications to the model architecture or model hyperparameters.

## Semi-supervised learning with pseudo-labels using Monte Carlo quality assessment (MCQA)

Although AdaBN facilitates domain transfer, it alone cannot completely solve the challenging problem of transfer learning between species, as the gap between different species lies not only in the MRI intensities but also in the brain structure and geometry. In this case, additional supervision is still necessary to guide the model to achieve highly accurate brain segmentation in the new species. Instead of common approaches such as training from scratch for fine-tuning, we propose a novel semi-supervised learning strategy for BEN. In addition to sparsely labeled data in the target domain, we make use of abundant unlabeled data by means of an iterative pseudo-labeling process (*Figure 2D*): we first evaluate the model on all unlabeled data and then select predictions that are associated with the lowest uncertainty (highest confidence) based on MCQA (detailed in subsequent sections). The predictions selected through quality screening are taken as pseudo-labels and are added to the training set in the next iteration. More specifically, we use a hybrid loss function:

$$\frac{1}{N} \sum_{i=1}^{n} \left[ \|f\left(x_i; \theta_t\right) - y_i\| + \lambda_{Dice} L_{dice} \left(f\left(x_i; \theta_t\right), y_i\right) \right] + \alpha\left(t\right) \frac{1}{M} \sum_{i=1}^{m} L_{dice} \left(f\left(x_j; \theta_t\right), f\left(x_j; \theta_{t-1}|\zeta\right)\right) \quad (1)$$

$$\alpha\left(t\right) = \begin{cases} 0, t < T_{sl} \\ \lambda_{sl}, t \geq T_{sl} \end{cases} \tag{2}$$

where $x_i$ represents a labeled data point with the ground-truth annotation $y_i$ in the current mini-batch (total number N) and $x_j$ denotes a pseudo-labeled data point whose model prediction $f\left(x_j; \theta_{t-1}\right)$ from the previous iteration is below an uncertainty threshold $\varsigma$ based on MCQA (total number M). $\alpha\left(t\right)$ is an important parameter that balances the relative contributions of true labels and pseudo-labels in the algorithm: we set this parameter to zero in the first several iterations, as our initial model is not good enough to make confident predictions, and then to a hyperparameter $\lambda$ from iteration $T_{sl}$ onward. The whole semi-supervised training procedure can be summarized as follows: The total number of training epochs is denoted by $T_{epoch}$, and the current epoch is denoted by $t$. During the first phase $(0 < t < T_{sl})$, the network is trained directly on the target domain using labeled data. Then, in the next phase $(T_{sl} \leq t \leq T_{epoch})$, unlabeled data with pseudo-labels additionally participate in the semi-supervised learning process by contributing to the loss function as determined by the balance coefficient $\alpha\left(t\right)$. Empirically, we set $T_{sl} = 20$, $\lambda_{Dice} = 1$, $\lambda_{sl} = 0.5$, and $\varsigma$ is the minimal uncertainty value in each minibatch.

## Model training implementation

The model was trained with the Dice loss function using the adaptive moment estimation (Adam) optimizer for 50 epochs with a batch size of 16 and learning rates of $10^{-4}$ and $10^{-5}$ in the source domain and all target domains, respectively. For training and testing on each domain, a separate cross-validation process was performed to assess the final performance of our method, with progressively larger numbers of observations (n=1, 2, 3,…) assigned to the training set while the rest of the samples were assigned to the test set. The mechanisms for domain transfer and semi-supervised learning have been described in previous sections.

## Model input preprocessing

To lessen the variation across individual scans, we applied the following processes on all datasets. First, we applied the N4 algorithm (*Tustison et al., 2010*) to remove bias fields in the MRI scans. Second, we normalized each image as follows (with $V_{norm}$ denoting the normalized MRI volume and $V^1$ and $V^{99}$ denoting the 1st and 99th percentiles, respectively, of the intensity values in the input volumes):

$$V_{norm} = \frac{V - V^1_{min}}{V^{99}_{max} - V^1_{min}} \tag{3}$$

For all functional MRI data, our model can process each time-point image and produce a corresponding binary mask. For consistency with the structural modalities, we used the results computed for the 5th time point to calculate the performance metrics.

The processed volumes were then fed into the network as input, slice by slice, after being resized or cropped to a matrix with dimensions of 256*256 for compatibility with the network. In addition, data augmentation was employed to empower the network to learn invariant characteristics from data exhibiting variations among different subjects and protocols. For each input slice, the following transformations were implemented: random rotation (±10), random scaling (90–110%) and random translation (up to ± 10 pixels along each axis). For example, the rotation and translation operators can imitate conditions in which the subjects are represented in diverse coordinate systems and have different head orientations. For the case of inconsistent voxel sizes across different sites and even for the lifespan of one individual during brain maturation, scaling can simulate corresponding image data.

## Model output postprocessing for uncertainty estimation and quality assessment

To understand the intrarater divergences as represented in BEN and provide valuable insights into model interpretability for clinicians, we utilize dropout layers in the network to obtain uncertainty measures for the model-generated segmentations.

There are two principal types of prediction uncertainties for deep neural networks: aleatoric uncertainty and epistemic uncertainty (*Hüllermeier and Waegeman, 2021*; *Kwon et al., 2020*). Aleatoric

uncertainty captures the potential inherent noise of the input test data, and therefore, this type of uncertainty is unlikely to be lessened by increasing the amount of training data. In contrast, epistemic uncertainty refers to the uncertainty in the model parameters, which represents the lack of knowledge of the best model. In DL networks, epistemic uncertainty can be caused by a lack of training data in certain areas of the input domain and will decrease with increased diversity of the training data distribution.

In our study, approximate Bayesian inference was used to estimate uncertainty. Specifically, keeping the dropout layers active in the inference phase, we collected N independent samples for each subject. The pixelwise uncertainties of each sample are given by the following formula:

$$\text{Uncertainty} = \underbrace{\sqrt{\frac{1}{T}\sum_{t=1}^{T} p_t \left(1 - p_t\right)}}_{\text{aleatoric}} + \underbrace{\sqrt{\frac{1}{T}\sum_{t=1}^{T} \left(p_t - \bar{p}\right)^2}}_{\text{epistemic}} \tag{4}$$

where $p_t$ denotes the sigmoid probability output in the final layer of the network in the inference stage and $\bar{p}$ is the average prediction for all inferences. In this work, we chose $T = 10$, which offers a good trade-off between the reliability of uncertainty estimation and the time consumed for Monte Carlo sampling, except in the human expert disagreement study, for which we set $T = 7$ for consistency with the number of clinical raters.

Based on these estimated uncertainty observations, we can filter out relatively unreliable predictions (usually those with high uncertainty values) based on the guidance obtained through statistical analysis, thereby realizing quality assessment. Moreover, relatively convincing predictions can be further utilized as pseudo-labels to guide the semi-supervised learning process.

## Baseline methods and traditional toolbox settings

To demonstrate the utility of domain transfer, we additionally tested two baseline methods of training the BEN model: training from scratch and fine-tuning. These methods share the same network architecture, learning rate, batch size and number of training epochs. For training from scratch, the weights of the layers in the network are randomly initialized, and then all weights are adjusted based on the training set. In the second baseline method, the training process is initialized on the source domain, and the weights are then fine-tuned to the new task in accordance with the target-domain dataset.

We also compared BEN with four widely used tools FSL (*Jenkinson et al., 2012*), AFNI (*Cox, 2012*), FreeSurfer (*Fischl, 2012*), and Sherm (*Liu et al., 2020*) representing SOTA processing pipelines in their respective research fields. We adjusted the parameters provided by these traditional toolboxes to adapt to the different domain datasets to achieve better performance; however, for situations in which the demands were beyond the scope of the capability of the tools, the default parameters were used instead. The detail parameter settings for these four methods are shown in *Appendix 1—table 7*.

## Ground-truth generation

The ground-truth brain and nonbrain regions for each domain dataset were manually annotated by two experts in the field of brain anatomy. Discrepancies were addressed through consensus. In addition, the labels for the macaque dataset were provided by a previous study (*Wang et al., 2021*), and we made no further modifications. The human brain masks of the ABCD, UKB, and ZIB datasets used in this work were obtained from efforts and contributions reported in previous literature (*Peng et al., 2021*; *Alfaro-Almagro et al., 2018*; *Casey et al., 2018*; *Gong et al., 2021*; *Miller et al., 2016*). All of the labels have been visually inspected.

## Evaluation metrics

The following two metrics are used to quantitatively compare the segmentation performance. The Dice score measures the overlap between the segmentation results and the ground truth. The definition of this measure is formulated as follows:

$$Dice = \frac{2|S \cap G|}{|S| + |G|}$$

where $S$ and $G$ denote the predicted and ground-truth label maps, respectively. Higher scores indicate better consistency. The 95% Hausdorff distance (HD95) measures how far the segmentation results and ground truth for a metric space are from each other as given by the following equation:

$$\text{HD}_{95}\left(S,G\right) = max\left(E^{95th}_{s\in\partial(S)}\min_{g\in\partial(G)}\left\{\|s-g\|\right\}, E^{95th}_{g\in\partial(G)}\min_{s\in\partial(S)}\left\{\|s-g\|\right\}\right)$$

where $\partial\left(\cdot\right)$ represents the boundary of the set and $E^{95th}_{s\in\partial(S)}$ and $E^{95th}_{g\in\partial(G)}$ are the 95th percentiles of the Euclidean distances between pixels in $S$ and $G$, respectively.

## Statistics

Statistical analyses were performed using SciPy (version 1.3.3) in Python (version 3.6.10).

The exact sample values and results of statistical tests, including the Mann–Whitney test and paired t-test, are provided in the figure captions or within the figure panels themselves. Differences are considered statistically significant when p<0.05.

## Code availability

We release BEN and pretrained models via https://github.com/yu02019/BEN (*Yu, 2023*).

## Acknowledgements

This work was supported by the Shanghai Municipal Science and Technology Major Project (No.2018SHZDZX01), ZJLab, Shanghai Center for Brain Science and Brain-Inspired Technology, National Natural Science Foundation of China (81873893, 82171903, 92043301), the Office of Global Partnerships (Key Projects Development Fund) at Fudan University.

---

## Additional information

### Competing interests

Dinggang Shen: is affiliated with Shanghai United Imaging Intelligence Co., Ltd. He has financial interests to declare. The other authors declare that no competing interests exist.

### Funding

| Funder | Grant reference number | Author |
| --- | --- | --- |
| National Natural Science Foundation of China | 82171903 | Xiao-Yong Zhang |
| Fudan University | the Office of Global Partnerships (Key Projects Development Fund) | Xiao-Yong Zhang |
| Shanghai Municipal Science and Technology Major Project | No.2018SHZDZX01 | Xiao-Yong Zhang Jianfeng Feng |
| National Natural Science Foundation of China | 81873893 | Xiao-Yong Zhang |
| National Natural Science Foundation of China | 92043301 | Xiao-Yong Zhang |

The funders had no role in study design, data collection and interpretation, or the decision to submit the work for publication.

### Author contributions

Ziqi Yu, Conceptualization, Software, Formal analysis, Visualization, Methodology, Writing – original draft; Xiaoyang Han, Data curation, Formal analysis, Validation; Wenjing Xu, Resources, Data curation, Investigation; Jie Zhang, Resources, Supervision; Carsten Marr, Writing – review and editing;

Dinggang Shen, Supervision, Writing – review and editing; Tingying Peng, Software, Supervision, Methodology, Writing – review and editing; Xiao-Yong Zhang, Conceptualization, Resources, Supervision, Funding acquisition, Writing – original draft, Project administration, Writing – review and editing; Jianfeng Feng, Resources, Supervision, Funding acquisition

### Author ORCIDs
Ziqi Yu http://orcid.org/0000-0001-8201-5481
Xiaoyang Han http://orcid.org/0000-0002-3007-6079
Carsten Marr http://orcid.org/0000-0003-2154-4552
Xiao-Yong Zhang http://orcid.org/0000-0001-8965-1077
Jianfeng Feng http://orcid.org/0000-0001-5987-2258

### Ethics

The Zhangjiang International Brain Biobank (ZIB) protocols were approved by the Ethics Committee of Fudan University (AF/SC-03/20200722) and written informed consents were obtained from all volunteers. UK Biobank (UKB) and Adolescent Brain Cognitive Development (ABCD) are publicly available.

Partial rodent MRI data collection were approved by the Animal Care and Use Committee of Fudan University, China. The rest rodent data (Rat-T2WI-9.4T and Rat-EPI-9.4T datasets) are publicly available (CARMI: https://openneuro.org/datasets/ds002870/versions/1.0.0). Marmoset MRI data collection were approved by the Animal Care and Use Committee of the Institute of Neuroscience, Chinese Academy of Sciences, China. Macaque MRI data are publicly available from the nonhuman PRIMatE Data Exchange (PRIME-DE) (https://fcon_1000.projects.nitrc.org/indi/indiPRIME.html).

### Decision letter and Author response

Decision letter https://doi.org/10.7554/eLife.81217.sa1
Author response https://doi.org/10.7554/eLife.81217.sa2

## Additional files

### Supplementary files

- MDAR checklist

### Data availability

We release a longitudinal MRI dataset of young adult C57BL6J mouse via https://zenodo.org/record/6844489. The rest rodent data (Rat-T2WI-9.4T and Rat-EPI-9.4T datasets) are publicly available (CARMI: https://openneuro.org/datasets/ds002870/versions/1.0.0). Macaque MRI data are publicly available from the nonhuman PRIMatE Data Exchange (PRIME-DE) (https://fcon_1000.projects.nitrc.org/indi/indiPRIME.html; *Milham et al., 2018*). UK Biobank (UKB) and Adolescent Brain Cognitive Development (ABCD) are publicly available. All code used in this work is released via https://github.com/yu02019/BEN (*Yu, 2023*).

The following dataset was generated:

| Author(s) | Year | Dataset title | Dataset URL | Database and Identifier |
| --- | --- | --- | --- | --- |
| Yu Ziqi, Xu W, Zhang X-Y | 2022 | A longitudinal MRI dataset of young adult C57BL6J mouse brain | https://doi.org/10.5281/zenodo.6844489 | Zenodo, 10.5281/zenodo.6844489 |

The following previously published datasets were used:

| Author(s) | Year | Dataset title | Dataset URL | Database and Identifier |
|---|---|---|---|---|
| Hsu L-M, Ban W, Chao T-H, Song S, Cerri DH, Walton L, Broadwater M, Lee S-H, Shih YI | 2020 | CAMRI Rat Brain MRI Data | https://doi.org/10.18112/openneuro.ds002870.v1.0.1 | OpenNeuro, 10.18112/openneuro.ds002870.v1.0.1 |
| Hsu L-M, Ban W, Chao T-H, Song S, Cerri DH, Walton L, Broadwater M, Lee S-H, Shih YI | 2020 | CAMRI Mouse Brain MRI Data | https://doi.org/10.18112/openneuro.ds002868.v1.0.1 | OpenNeuro, 10.18112/openneuro.ds002868.v1.0.1 |

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

## Appendix 1

**Appendix 1—table 2.** Performance comparison of BEN with SOTA methods on the source domain (Mouse-T2WI-11.7T).
Dice: Dice score; SEN: sensitivity; SPE: specificity; ASD: Average Surface Distance; HD95: the 95-th percentile of Hausdorff distance.

| Method | Dice | SEN | SPE | ASD | HD95 |
|---|---|---|---|---|---|
| Sherm | 0.9605 | 0.9391 | **0.9982** | 0.6748 | 0.4040 |
| AFNI | 0.9093 | 0.9162 | 0.9894 | 1.9346 | 0.9674 |
| FSL | 0.3948 | **1.0000** | 0.6704 | 20.4724 | 5.5975 |
| BEN | **0.9859** | 0.9889 | **0.9982** | **0.3260** | **0.1436** |

**Appendix 1—table 3.** Performance comparison of BEN with SOTA methods on two public datasets.
Dice: Dice score; Jaccard: Jaccard Similarity; SEN: sensitivity; HD: Hausdorff distance.

CARMI dataset * (Rodent)

T2WI

| Methods | Dice | Jaccard | SEN | HD (voxels) |
|---|---|---|---|---|
| RATS (*Oguz et al., 2014*) | 0.91 | 0.83 | 0.85 | 8.76 |
| PCNN (*Chou et al., 2011*) | 0.89 | 0.80 | 0.90 | 7.00 |
| SHERM (*Liu et al., 2020*) | 0.88 | 0.79 | 0.86 | 6.72 |
| U-Net (*Hsu et al., 2020*) | 0.97 | 0.94 | 0.96 | 4.27 |
| BEN | **0.98** | **0.95** | **0.98** | **2.72** |

EPI

| Methods | Dice | Jaccard | SEN | HD (voxels) |
|---|---|---|---|---|
| RATS (*Oguz et al., 2014*) | 0.86 | 0.75 | 0.75 | 7.68 |
| PCNN (*Chou et al., 2011*) | 0.85 | 0.74 | 0.93 | 8.25 |
| SHERM (*Liu et al., 2020*) | 0.80 | 0.67 | 0.78 | 7.14 |
| U-Net (*Hsu et al., 2020*) | 0.96 | 0.93 | 0.96 | 4.60 |
| BEN | **0.97** | **0.94** | **0.98** | **4.20** |

PRIME-DE † (Macaque)

T1WI

| Methods | Dice | Jaccard | SEN | HD (voxels) |
|---|---|---|---|---|
| FSL | 0.81 | 0.71 | 0.96 | 32.38 |
| FreeSurfer | 0.56 | 0.39 | **0.99** | 42.18 |
| AFNI | 0.86 | 0.79 | 0.82 | 25.46 |
| U-Net (*Wang et al., 2021*) | 0.98 | - | - | - |
| BEN | **0.98** | 0.94 | 0.98 | **13.21** |

*https://openneuro.org/datasets/ds002870/versions/1.0.0.
†https://fcon_1000.projects.nitrc.org/indi/indiPRIME.html.

**Appendix 1—table 4.** Ablation study of each module of BEN in the source domain.
(a) training with all labeled data using U-Net. The backbone of BEN is non-local U-Net (NL-U-Net).

**Appendix 1—table 1.** MRI scan information of the fifteen animal datasets and three human datasets.
FDU: Fudan University. UCAS: University of Chinese Academy of Sciences; UNC: University of North Carolina at Chapel Hill; ABCD: Adolescent Brain Cognitive Developmental study; ZIB: Zhangjiang International Brain BioBank at Fudan University.

| Species | Modality | Magnetic Field (T) | Scans | Slices | In-plane Resolution (mm) | Thickness (mm) | Manufacturer | Institution |
|---|---|---|---|---|---|---|---|---|
| Mouse | T2WI | 11.7 | 243 | 9,030 | 0.10*0.10 | 0.4 | Bruker | FDU |
| | | 9.4 | 14 | 448 | 0.06*0.06 | 0.4 | Bruker | UCAS |
| | | 7 | 14 | 126 | 0.08*0.08 | 0.8 | Bruker | UCAS |
| | EPI | 11.7 | 54 | 2,198 | 0.2*0.2 | 0.4 | Bruker | FDU |
| | | 9.4 | 20 | 360 | 0.15*0.15 | 0.5 | Bruker | UCAS |
| | SWI | 11.7 | 50 | 1,500 | 0.06*0.06 | 0.5 | Bruker | FDU |
| | ASL | 11.7 | 58 | 696 | 0.167*0.167 | 1 | Bruker | FDU |
| Rat | T2WI | 11.7 | 132 | 5,544 | 0.14*0.14 | 0.6 | Bruker | FDU |
| | | 9.4 | 55 | 660 | 0.1*0.1 | 1 | Bruker | UNC † |
| | | 7 | 88 | 4,400 | 0.09*0.09 | 0.4 | Bruker | UCAS |
| | EPI | 9.4 | 55 | 660 | 0.32*0.32 | 1 | Bruker | UNC † |
| **Sum of rodent** | | | 783 | 25,622 | | | | |
| Marmoset | T2WI | 9.4 | 62 | 2,480 | 0.2*0.2 | 1 | Bruker | UCAS |
| | EPI | 9.4 | 50 | 1,580 | 0.5*0.5 | 1 | Bruker | UCAS |
| Macaque * | T1WI | 4.7 3 1.5 | 76 | 20,063 | 0.3*0.3~0.6*0.6 | 0.3~0.75 | Siemens, Bruker, Philips | Multicenter |
| | EPI | 1.5 | 58 | 2,557 | 0.7*0.7~2.0*2.0 | 1.0~3.0 | Siemens, Bruker, Philips | Multicenter |
| **Sum of nonhuman primate** | | | 246 | 26,680 | | | | |

*Appendix 1—table 1 continued on next page*

Appendix 1—table 1 continued

| Species | Modality | Magnetic Field (T) | Scans | Slices | In-plane Resolution (mm) | Thickness (mm) | Manufacturer | Institution |
|---|---|---|---|---|---|---|---|---|
| Human (ABCD) | T1WI | 3 | 3,250 | 552,500 | 1.0*1.0 | 1 | GE Siemens Philips | Multicenter |
| Human (UK Biobank) | T1WI | 3 | 963 | 196,793 | 1.0*1.0 | 1 | Siemens | Multicenter |
| Human (ZIB) | T1WI | 3 | 388 | 124,160 | 0.8*0.8 | 0.8 | Siemens | FDU |
| Sum of human | | | 4,601 | 873,453 | | | | |
| In total | | | 5,630 | 925,755 | | | | |

*https://fcon_1000.projects.nitrc.org/indi/indiPRIME.html.
†https://openneuro.org/datasets/ds002870/versions/1.0.0.

**Appendix 1—table 7.** Protocols and parameters used for conventional neuroimaging toolboxes.

| Method | Command | Parameter | Description | Range | Chosen value |
|---|---|---|---|---|---|
| AFNI | 3dSkullStrip | -marmoset | Brain of a marmoset | on/off | on for marmoset |
| | | -rat | Brain of a rat | on/off | on for rodent |
| | | -monkey | Brain of a monkey | on/off | on for macaque |
| FreeSurfer | mri_watershed | -T1 | Specify T1 input volume | on/off | on |
| | | -r | Specify the radius of the brain (in voxel unit) | positive number | 60 |
| | | -less | Shrink the surface | on/off | off |
| | | -more | Expand the surface | on/off | off |
| FSL | bet2 | -f | Fractional intensity threshold | 0.1~0.9 | 0.5 |
| | | -m | Generate binary brain mask | on/off | on |
| | | -n | Don't generate the default brain image output | on/off | on |
| Sherm | sherm | -animal | Species of the task | 'rat' or 'mouse' | according to the task |
| | | -isotropic | Characteristics of voxels | 0/1 | 0 |

(b) training with 5% labeled data. (c) training with 5% labeled data using BEN's semi-supervised learning module (SSL). The remaining 95% of the unlabeled data is also used for the training. Since this ablation study is performed on the source domain, the adaptive batch normalization (AdaBN) module is not used. Dice: Dice score; SEN: sensitivity; SPE: specificity; HD95: the 95-th percentile of Hausdorff distance.

| Method | Scans used | | Metrics | | | |
|---|---|---|---|---|---|---|
| | Labeled | Unlabeled | Dice | SEN | SPE | HD95 |
| [a]U-Net | 243 | 0 | 0.9773 | 0.9696 | **0.9984** | 0.2132 |
| [a]Backbone | 243 | 0 | **0.9844** | **0.9830** | **0.9984** | **0.0958** |
| [b]U-Net | 12 | 0 | 0.9588 | 0.9546 | 0.9945 | 1.1388 |
| [b]Backbone | 12 | 0 | 0.9614 | 0.9679 | **0.9970** | 0.7468 |
| [c]Backbone +SSL | 12 | 231 | **0.9728** | **0.9875** | 0.9952 | **0.2937** |

**Appendix 1—table 5.** Ablation study of each module of BEN in the target domain.
(a) training from scratch with all labeled data. The backbone of BEN is non-local U-Net (NL-U-Net). (b) training from scratch with 5% labeled data. (c) fine-tuning (using pretrained weights) with 5% labeled data. (d) fine-tuning with 5% labeled data using BEN's SSL and AdaBN modules. The remaining 95% of the unlabeled data is also used for the training stage. Dice: Dice score; SEN: sensitivity; SPE: specificity; HD95: the 95-th percentile of Hausdorff distance.

| Method | Pretrained | Scans used | | Metrics | | | |
|---|---|---|---|---|---|---|---|
| | | Labeled | Unlabeled | Dice | SEN | SPE | HD95 |
| [a]Backbone (from scratch) | | 132 | 0 | **0.9827** | 0.9841 | 0.9987 | **0.1881** |
| [b]Backbone (from scratch) | | 7 | 0 | 0.8990 | 0.8654 | 0.9960 | 4.6241 |
| [c]Backbone | ✓ | 7 | 0 | 0.9483 | 0.9063 | **0.9997** | 0.6563 |
| [c]Backbone +AdaBN | ✓ | 7 | 0 | 0.9728 | **0.9875** | 0.9952 | 0.2937 |
| [d]Backbone +SSL | ✓ | 7 | 125 | 0.9614 | 0.9679 | 0.9970 | 0.7468 |
| [d]Backbone +AdaBN + SSL | ✓ | 7 | 125 | **0.9779** | 0.9763 | 0.9986 | **0.2912** |

**Appendix 1—table 6.** BEN provides interfaces for the following conventional neuroimaging software.

| Name | Link |
|---|---|
| AFNI | https://afni.nimh.nih.gov/ |
| ANTs | http://stnava.github.io/ANTs/ |
| FSL | https://fsl.fmrib.ox.ac.uk/fsl/fslwiki |
| FreeSurfer | https://freesurfer.net/ |
| SPM | http://www.fil.ion.ucl.ac.uk/spm |
| Nipype | https://pypi.org/project/nipype/ |

