## [Editor Report]

This article is an important contribution to the field of neuroimaging. The paper proposes a deep neural network for brain extraction and an approach to training the network that generalises across domains, including species, scanners, and MRI sequences. The authors provide convincing evidence that their approach works for a varied set of data, protocols, and species.

---

## [Decision Letter]

**Decision letter after peer review:**

Thank you for submitting your article "A generalizable brain extraction net (BEN) for multimodal MRI data from rodents, nonhuman primates, and humans" for consideration by *eLife*. Your article has been reviewed by 2 peer reviewers, and the evaluation has been overseen by a Reviewing Editor and Floris de Lange as the Senior Editor. The following individuals involved in the review of your submission have agreed to reveal their identity: Emma C. Robinson, PhD (Reviewer #1); Jason P Lerch (Reviewer #2).

Essential revisions:

The one major point raised by reviewer 2 appears to me to be the most important to properly address, as it appears the method did not work well on the reviewer's own data, casting doubt on the generalisability of the approach- the main selling point of the paper.

*Reviewer #1 (Recommendations for the authors):*

I recommend that the paper is largely ready for publication in its current form.

*Reviewer #2 (Recommendations for the authors):*

The major point I'd like to see the authors discuss is when BEN needs to be retrained on different input data and discuss techniques to improve generalizability. In the examples given and weights provided they suggest that, for example, 7T and 9.4T T2w mouse data needs different networks. This is somewhat surprising to me and suggests that the networks might be overfitting to their input data. My own tests (as described in the public review) also suggest that even subtle changes to out-of-sample data quickly degrade performance.

Secondly, I find that the narrative in places overstates the importance of their work, primarily since in my opinion the community has created multiple brain masking algorithms in different species that work well. Three examples include:

1) Line 17: the claim that brain extraction in animals is not fully automated; the relatively simpler brains, especially in rodents, means that image registration-based approaches to segmenting brains is quite successful and has been implemented in multiple toolkits. Similarly, the claim that the performance of registration-based methods is limited is at odds with the data.

2) It is not clear to me why the authors would expect FSL or FreeSurfer to work on rodents out of the box, given that the algorithms were never tuned for non-human brains (as far as I am aware). Their inclusion for animal brain segmentation tasks thus appears to be a bit of a straw man.

3) I also found Figure 7 and the related arguments about why BEN is necessary a bit odd; any decent registration/segmentation pipeline would incorporate brain masking, so the comparison of with and without masking is also a false contrast. There are lots of interesting ideas in this manuscript that it does not need these types of strawman arguments, so I would suggest removing this section entirely or alternately comparing the inclusion of BEN for masking as compared to alternate pipelines with masking included as well.

---

## [Author Response]

Essential revisions:The one major point raised by reviewer 2 appears to me to be the most important to properly address, as it appears the method did not work well on the reviewer's own data, casting doubt on the generalisability of the approach- the main selling point of the paper.

We thank the Editors very much. These comments are very encouraging, valuable, and constructive. We have carefully revised our manuscript based on the comments of the reviewers. Please refer to our response to the reviewers for details.

Reviewer #2 (Recommendations for the authors):The major point I'd like to see the authors discuss is when BEN needs to be retrained on different input data and discuss techniques to improve generalizability. In the examples given and weights provided they suggest that, for example, 7T and 9.4T T2w mouse data needs different networks. This is somewhat surprising to me and suggests that the networks might be overfitting to their input data. My own tests (as described in the public review) also suggest that even subtle changes to out-of-sample data quickly degrade performance.

We thank the reviewer for the helpful comment and raise important aspects we have addressed in our paper and Github codes. The adjustments are listed as follows:

1. We have discussed briefly when BEN needs to be retrained in the second paragraph in “Discussion – Compatibility with other toolboxes and deployment”. In the revised version, we have updated the descriptions to make it clear, and also provide several video tutorials (using public ex vivo data to demonstrate). Since the initial version of BEN is intended to reproduce our results in paper, some corner cases and complex cases were not well taken into consideration. These concerns have now been well addressed in updated codes (https://github.com/yu02019/BEN).

2. The techniques for increasing generalizability have also been added to “Discussion” and BEN pipeline, e.g., orientation detection, post-processing, etc.

3. As for experiments across 7T and 9.4T T2w, BEN could adapt to these out-of-domain images without labels (zero-shot learning) , as the domain shift between 7T and 9.4T is relatively small. The quantitative results for these cross-MR scanners with various magnetic field strengths are presented in Figure 3—figure supplement 2. Without additional labels (zero-shot), BEN presents satisfactory performance on two of the three tasks, while the other two baseline methods all fail on all three tasks. In fact, we deem this to be essentially a cross-center task that disables the performances of many deep learning-based methods; BEN addresses and alleviates this problem using the "domain adaptation module" and packages it well for the general user without coding skills. Alternatively, one can train a “super-network” with big training data across different species, modalities and magnetic strengths so the network can be used in an ‘out-of-the-box' fashion. But this is difficult in practice as the animal MR experiments are very diverse in nature and there is always some “out-of-sample” testing data.

4. Besides, the original intent for cross-field strength experiments was to demonstrate BEN could provide a fast and label-efficient domain adaptation training method as a scalable tool. We can certainly provide joint-training weights that could easily meet the reviewer’s requirement. However, when deploying BEN or other toolboxes into customized data/cohorts, it is inevitable to address domain adaptation issues, suggesting the importance of our model design.

5. How to deal with out-of-sample data is a challenge for many deep learning methods. To the best of our knowledge, it is very difficult for almost all existing methods to handle the images with low quality (artifacts, field inhomogeneity, high noise, or low SNR). To solve this issue, in our model, we suggest adding several exemplary MR scans to retrain BEN; on the other hand, it could be addressed partly by post-processing or the “Monte Carlo quality assessment module” in the BEN pipeline. We will give some examples in our tutorials.

Secondly, I find that the narrative in places overstates the importance of their work, primarily since in my opinion the community has created multiple brain masking algorithms in different species that work well. Three examples include:1) Line 17: the claim that brain extraction in animals is not fully automated; the relatively simpler brains, especially in rodents, means that image registration-based approaches to segmenting brains is quite successful and has been implemented in multiple toolkits. Similarly, the claim that the performance of registration-based methods is limited is at odds with the data.

We thank the reviewer for raising this issue. We think the success of automatic registration-based approaches depends on the quality and the number of atlases (e.g. multi-atlas registration is usually better than single-atlas one), and registration is not an easy task as the heterogeneous contrasts existing in different image spaces (e.g., native space and atlas space) might not provide enough guidance for intensity-based registration metrics. Due to the scarcity of publicly available MRI data and multi-atlases, it is generally more difficult in animal MR studies than human ones where multi-atlas registration is well established.

Alternatively, some semi-automatic registration-based approaches using one dataset-specific template atlas which have to be manually labeled (then it has similar or identical experimental conditions, MR parameters, and image properties) present better and more stable performance on the current experimental cohort than fully automated methods using public atlas from different cohorts or imaging centers. There still remain limitations to semi-automatic registration-based methods: (1) If registration-based methods fail, it’s hard to adjust, thus requiring laborious manual corrections. (2) The dataset-specific template mask is usually manually annotated, which is another time-consuming step in addition to the registration. (3) Based on our results (Author response table 1), registration-based methods performed unsatisfactory on fMRI data and scans with thick slice thickness.

**Author response table 1. sa2table1:** Comparison of brain extraction performance of different methods on different datasets. SkullStrip: semi-automatic registration-based method. (ASD: average surface distance. HD95: 95% Hausdorff distance).

Species	Field	Modality	Method	Dice	Sensitivity	Specificity	ASD	HD95
Mouse	11.7T	T2WI	**BEN**	**0.9859**	0.9889	**0.9982**	**0.3260**	**0.1436**
			Sherm	0.9605	0.9391	0.9982	0.6748	0.4040
			AFNI	0.9093	0.9162	0.9894	1.9346	0.9674
			SkullStrip	0.9697	0.9783	0.9957	0.4729	0.2829
			FSL	0.3948	**1.0000**	0.6704	20.4724	5.5975
		EPI	**BEN**	**0.9791**	**0.9912**	0.9970	0.5945	0.4946
			Sherm	0.9440	0.9206	**0.9971**	**0.4827**	**0.4896**
			AFNI	0.9139	0.9365	0.9890	0.7835	0.7315
			SkullStrip	0.9237	0.9502	0.9899	0.8570	0.5673
		SWI	**BEN**	**0.9879**	**0.9912**	0.9979	**0.4548**	**0.2420**
			Sherm	0.9586	0.9342	**0.9983**	0.4633	0.4019
			AFNI	0.9060	0.8631	0.9950	0.8632	0.5522
			SkullStrip	0.9600	0.9590	0.9954	0.4777	0.3797
		ASL	**BEN**	0.**9807**	**0.9804**	**0.9966**	**0.1766**	**0.2679**
			Sherm	0	0	0.9938	3.5669	10.6817
			AFNI	0.7584	0.9379	0.9324	4.3252	2.3752
			SkullStrip	0.8893	0.9177	0.9737	0.7351	0.7029
	9.4T	T2WI	**BEN**	**0.9830**	**0.9903**	**0.9954**	**0.5129**	**0.3284**
			Sherm	0.8675	0.7893	0.9951	1.7973	1.4372
			AFNI	0.8699	0.9532	0.9548	2.2979	1.2606
			SkullStrip	0.9298	0.9683	0.9750	1.5024	0.8822
		EPI	**BEN**	**0.9535**	0.9567	**0.9920**	**0.5686**	0.5021
			Sherm	0.9368	0.9394	0.9901	0.5956	**0.4370**
			AFNI	0.8855	**0.9585**	0.9650	0.9960	0.6349
			SkullStrip	0.9249	0.9613	0.9810	0.7440	0.6718
	7T	T2WI	**BEN**	**0.9815**	**0.9682**	**0.9951**	**0.4272**	**0.2595**
			Sherm	0.8436	0.9423	0.9596	4.3723	1.6113
			AFNI	0.9123	0.9423	0.9910	2.4293	0.8190
			SkullStrip	0.8599	0.9423	0.9703	1.5989	0.9821
Rat	7T	T2WI	**BEN**	**0.9854**	**0.9878**	0.9966	**0.4688**	**0.1790**
			Sherm	0.9532	0.9314	0.9956	0.8428	1.0463
			AFNI	0.8285	0.7232	**0.9969**	2.8672	2.7561
			FSL	-	-	-	-	-
			SkullStrip	0.9754	0.9813	0.9939	0.5015	0.4158
		EPI	**BEN**	**0.9705**	0.9415	0.9966	**0.2858**	**0.7081**
			Sherm	0.6339	0.4645	**0.9998**	1.2283	3.4795
			AFNI	0.8080	0.7151	0.9896	0.9018	4.2521
			SkullStrip	0.9394	**0.9500**	0.9869	0.4161	0.7991
Marmoset	9.4T	T2WI	**BEN**	**0.9804**	0.9788	**0.9957**	0.5568	5.7393
			Sherm	-	-	-	-	-
			AFNI	0.9311	0.9526	0.9803	1.1744	9.7897
			SkullStrip	0.9755	**0.9789**	0.9943	**0.4005**	**3.8443**
			FSL	0.7689	0.8126	0.9552	2.0939	5.1353
		EPI	**BEN**	**0.9774**	**0.9816**	0.9949	0.9126	3.500
			Sherm	-	-	-	-	-
			AFNI	0.9159	0.9142	0.9847	0.8314	5.1391
			SkullStrip	0.9533	0.9286	**0.9965**	**0.4489**	**3.0342**
			FSL	0.9326	0.9748	0.9757	1.0846	8.7247

We have conducted comparisons using such semi-automatic registration-based approaches, “SkullStrip” [1] (based on NiftyReg). The results are presented in Author response table 1 as follows. As a conclusion, though [1] could perform well in some cases, it seems that its performance suffers from functional MR data and scans with thick slice thickness (e.g., Mouse-ASL-11.7T and Mouse-T2WI-9.4T images, etc.). Besides, BEN’s results show a much lower HD95 value, which means higher boundary agreement with the ground truth. One more point we want to emphasize is that the computational speed for such registration-based methods (which take hours for one small cohort) is much slower than BEN (which only takes several minutes).

Moreover, unlike registration-based methods (it’s hard to adjust when dealing with failed segmentations), BEN could improve its output by updating annotations of failed cases, which is like a “human-in-the-loop” manner. In addition, BEN‘s clear running logs and interactive training procedures would further provide more information for researchers.

[1] Delora A, Gonzales A, Medina C S, et al. A simple rapid process for semi-automated brain extraction from magnetic resonance images of the whole mouse head[J]. Journal of neuroscience methods, 2016, 257: 185-193.

2) It is not clear to me why the authors would expect FSL or FreeSurfer to work on rodents out of the box, given that the algorithms were never tuned for non-human brains (as far as I am aware). Their inclusion for animal brain segmentation tasks thus appears to be a bit of a straw man.

We thank the reviewer for this comment. We know that FSL/FreeSurfer are designed for human beings, and are not designed for rodents. Here we include these tools just for parallel comparison (otherwise we do not have enough comparison), not criticizing these well-established tools. Since such publicly available animal neuroimaging tools are scarce, we include these tools in our consideration of the influence and seminal role of these tools, and it’s impractical to find a more suitable substitute for each species and modality. Indeed, we focus on the comparison of BEN and FSL/FreeSurfer in human brain performance (Figure 4 E, J and O and Figure 5—figure supplement 3). As BEN could achieve competitive performance on human brains compared with FSL/FreeSurfer and the transferability of BEN made it not bound to specific species or modality, readers can easily deploy BEN on their customized dataset, without the requirement of complex programming skills or mathematics knowledge.

3) I also found Figure 7 and the related arguments about why BEN is necessary a bit odd; any decent registration/segmentation pipeline would incorporate brain masking, so the comparison of with and without masking is also a false contrast. There are lots of interesting ideas in this manuscript that it does not need these types of strawman arguments, so I would suggest removing this section entirely or alternately comparing the inclusion of BEN for masking as compared to alternate pipelines with masking included as well.

We thank the reviewer for raising this valuable point. We have moved Figure 7 into supporting materials (Figure 5—figure supplement 1) and revised it (BEN vs AFNI-rats) in the revised version. The reason why we did not remove this figure is that we think this figure is an important example application to show the impact of BEN on downstream analysis. We agree that for human studies, brain masking is already integrated in the standard MR brain image processing pipeline, e.g. FSL or Freesurfer. Yet in animal studies, there are no standardized pipelines for decent animal brain segmentation/registration. Therefore, it is essential to show the strength of BEN in improving downstream analysis and its potential to be incorporated into a recommended animal MR brain processing pipeline.